# Defining diurnal fluctuations in mouse choroid plexus and CSF at high molecular, spatial, and temporal resolution

Ryann M. Fame [1,12], Peter N. Kalugin [1,2,3], Boryana Petrova [1], Huixin Xu [1], Paul A. Soden[1], Frederick B. Shipley[1,4], Neil Dani[1], Bradford Grant[1], Aja Pragana[1], Joshua P. Head[1], Suhasini Gupta[1], Morgan L. Shannon[1], Fortunate F. Chifamba[5,6], Hannah Hawks-Mayer[5,6], Amanda Vernon[7,8,9], Fan Gao[7,8,9,13], Yong Zhang[10], Michael J. Holtzman [10], Myriam Heiman [7,8,9], Mark L. Andermann [2,4,11], Naama Kanarek [1,9], Jonathan O. Lipton [5,6] & Maria K. Lehtinen [1,2,4,9] ✉

Transmission and secretion of signals via the choroid plexus (ChP) brain barrier can modulate brain states via regulation of cerebrospinal fluid (CSF) composition. Here, we developed a platform to analyze diurnal variations in male mouse ChP and CSF. Ribosome profiling of ChP epithelial cells revealed diurnal translatome differences in metabolic machinery, secreted proteins, and barrier components. Using ChP and CSF metabolomics and blood-CSF barrier analyses, we observed diurnal changes in metabolites and cellular junctions. We then focused on transthyretin (TTR), a diurnally regulated thyroid hormone chaperone secreted by the ChP. Diurnal variation in ChP TTR depended on *Bmal1* clock gene expression. We achieved real-time tracking of CSF-TTR in awake *Ttr^mNeonGreen* mice via multi-day intracerebroventricular fiber photometry. Diurnal changes in ChP and CSF TTR levels correlated with CSF thyroid hormone levels. These datasets highlight an integrated platform for investigating diurnal control of brain states by the ChP and CSF.

Cerebrospinal fluid (CSF) plays critical roles in regulating the central nervous system (CNS) throughout life, including the distribution of essential health and growth-promoting factors and clearing of waste from the brain[1]. CSF production robustly varies across the 24-h light–dark cycle[2]. CSF distribution between brain parenchyma interstitial fluid, ventricles, and cervical lymph nodes also varies across the 24-h day[3–5]. Over a century of studies suggest that the CSF contains biomarkers for circadian rhythmicity and plays key roles in relaying the output of diurnal clocks to target brain tissues. Early studies showed that CSF can carry circadian cues for drowsiness[6,7] and satiety[8], and experimental models indicate that diffusible factors released from the suprachiasmatic nucleus (SCN) of the hypothalamus into the CSF can mediate the circadian rhythmicity of locomotion[9–13]. The brain's SCN "master clock" is entrained by daily light–dark cycles and in a day of 12 h of light and 12 h of dark, C57BL/6 and CD1 laboratory mice are primarily active in the first half of the dark phase. However, progress

[1]Department of Pathology, Boston Children's Hospital and Harvard Medical School, Boston, MA 02115, USA. [2]Graduate Program in Neuroscience, Harvard Medical School, Boston, MA 02115, USA. [3]Harvard/MIT MD-PhD Program, Harvard Medical School, Boston, MA 02115, USA. [4]Graduate Program in Biophysics, Harvard University, Cambridge, MA 02138, USA. [5]Department of Neurology and the F.M. Kirby Neurobiology Center, Boston Children's Hospital, Boston, MA 02115, USA. [6]Division of Sleep Medicine, Harvard Medical School, Boston, MA 02115, USA. [7]Department of Brain and Cognitive Sciences, MIT, Cambridge, MA, USA. [8]Picower Institute for Learning and Memory, Cambridge, MA, USA. [9]Broad Institute of MIT and Harvard, Cambridge, MA, USA. [10]Pulmonary and Critical Care Medicine, Department of Medicine, Washington University, St. Louis, MO 63110, USA. [11]Division of Endocrinology, Diabetes, and Metabolism, Department of Medicine, Beth Israel Deaconess Medical Center, Boston, MA 02115, USA. [12]Present address: Department of Neurosurgery, Stanford University, Stanford, CA 94305, USA. [13]Present address: Lyterian Therapeutics, South San Francisco, 94080 CA, USA. ✉e-mail: maria.lehtinen@childrens.harvard.edu

toward understanding diurnal CSF regulation has been hampered by a limited understanding of how tissues that contribute to CSF govern its composition and a lack of specialized tools for tracking these changes in vivo or in real time.

The choroid plexus (ChP) is a key source of CSF. The specialized ChP epithelial cells that directly modulate CSF contents also display cell-autonomous circadian rhythmicity in gene expression of core components of the molecular clock[14–17]. These molecular circadian clocks depend on transcriptional-translational feedback loops as well as epigenetic, translational, and post-translational mechanisms that interact to enable both a robust and responsive clock[18–21]. In mammals, the molecular clock is a negative feedback loop in which a heterodimer of the master circadian regulator BMAL1 and its partners CLOCK (or NPAS2) activate transcription of clock-controlled genes including Period (Per) and Cryptochrome (Cry) that code for repressors of BMAL1 heterodimer activity, thus closing the loop that generates rhythms of approximately 24 h[22,23]. In addition to transcription, ribosome biogenesis and translation have emerged as direct outputs of the circadian molecular clock at the cellular level in synchronized cell lines and in the liver[24–26]. In fact, BMAL1 directly associates with translational machinery to promote protein synthesis[25]. While the circadian adaptations of each tissue involve both transcriptional and translational regulation, essentially nothing is known about the regulation of translation at the blood-CSF barrier.

Here, we developed an experimental platform to analyze the rhythmicity of ChP and CSF with respect to circadian clock gene expression and its dependence on external cues. We adapted tools to evaluate broad diurnal changes in transcript abundance, transcript ribosomal association, and protein levels. Using these tools, we pinpointed key pathways that change throughout the day and generated baseline datasets from unique, limited tissue to describe diurnal variations in the ChP translatome, secretome, and metabolome. These observations demonstrated widespread diurnal regulation of ChP secretion and barrier function across hours during the circadian day and in response to feeding cues, resulting in changes in CSF composition. We then established an intravital CSF fiber photometry system to track ChP output. Together, these multi-modal results illustrate the utility of our integrated platform for tracking diurnal dynamics and functions of the ChP-CSF system and raise testable hypotheses about how the ChP-CSF axis forms an important bridge between body and brain rhythmicity.

## Results

### Choroid plexus translation is diurnally regulated

To investigate whether translation and protein synthesis differ in ChP between light and dark phases, we monitored the mTOR-effector kinase ribosomal protein S6 kinase 1 (S6K1), which drives translation when phosphorylated. Using phosphorylation of the S6K target ribosomal protein S6 as a measure of mTOR pathway activity and general translation capacity, we observed increased phosphorylation of ribosomal protein S6 on Ser 240/244 (pS6) at 9 p.m. during the dark phase compared to 9 a.m. during the light phase (Fig. 1a–c). Higher pS6 at 9 p.m. suggests that increased levels of active translation machinery, and therefore higher protein synthesis capacity, occur during the dark phase relative to the light phase in mouse ChP. The circadian rhythmicity of ChP core molecular clock component transcripts Bmal1 and Per2 was in phase with that of the liver (Fig. 1d, Supplementary Fig. 1a–e). The liver served as an internal control since circadian changes in translation and metabolism have been robustly defined in the liver. The times of day with differential ChP pS6 expression (low at 9 a.m. and high at 9 p.m.) were inversely correlated with phasic Bmal1 mRNA expression. The relative increase in dark phase pS6 depended on Bmal1, as differential S6 phosphorylation was abrogated in ChP from Bmal1-knockout mice (Fig. 1e, f). Using ribosome biogenesis as another readout of increased ChP translation capacity, we found that

nucleolar volume as detected by the nucleolar protein Fibrillarin was higher at 9 p.m. than at 9 a.m. (Fig. 1g, h), consistent with reports in liver that the circadian clock coordinates ribosome biogenesis[24]. We next used O-propargyl-puromycin (OPP)[27] delivered by intraperitoneal injection to visualize actively elongating nascent polypeptides in vivo at the cellular level. OPP incorporation was higher at 9 p.m. compared to 9 a.m. in ChP epithelial cells (Fig. 1i), consistent with their larger nucleolar volumes and higher pS6 levels at night. Overall, translation was increased during the dark phase and diurnal variation in translation was dependent on systemic Bmal1 expression.

We next used ribosome profiling to determine whether overall levels of translation in ChP and the number of unique transcripts associated with ribosomes were increased during the dark phase (9 p.m.). Using Translating Ribosomal Affinity Purification (TRAP)[28,29] from ChP at 9 a.m. and 9 p.m., we identified transcripts that were prioritized for translation by the subset of polysome-enriched ribosomes containing large ribosomal subunit protein RPL10A[30]. ChP epithelial cells were targeted by crossing FoxJ1:cre mice[31] with floxed TRAP (EGFP:L10a) mice[29,32] (Fig. 1j), and mRNA associated with the large ribosomal subunit RPL10A was purified for sequencing from dissected lateral ventricle (LV) ChP (cre-negative animals were used as controls). TRAP analyses revealed unique transcripts differentially associated with RPL10A at 9 a.m. vs. 9 p.m. and those transcripts preferentially in the unbound supernatant (Fig. 1k). While both times had more unique transcripts in the supernatant than associated with RPL10A (Fig. 1l), both the total number of unique transcripts and the normalized number of reads (FPKM; Fragments Per Kilobase of transcript per Million mapped reads) of those transcripts associated with the ribosomes were higher during the dark phase (9 p.m.), consistent with a more diverse translatome and larger overall translation output at 9 p.m. during the dark phase in LV ChP than during the light phase at 9 a.m. (Fig. 1l, m). Together, this analysis of ChP translation and protein synthesis revealed meaningful differences between the dark and light phases including higher levels of pS6 and protein synthesis during the dark phase. Additionally, we demonstrated that Bmal1 was required for diurnal differences in phosphorylation of the ribosomal protein S6 in ChP (Fig. 1).

Next, we tested the differential association of specific transcripts with the ribosome between light and dark phases in ChP epithelial cells. Analyses of all ChP epithelial cell transcripts associated with ribosomes revealed 779 differentially translated transcripts at 9 a.m. vs. 9 p.m.: 431 enriched at 9 a.m., and 348 enriched at 9 p.m. (Fig. 2a). Gene sets identified from differentially enriched genes were substantially different between the two times (Fig. 2b–d; Supplementary Fig. 2a). Relevant to the role of ChP as a secretory epithelium, secreted proteins (Fig. 2e) and signal peptide-containing transcripts without a transmembrane domain, were differentially diurnally regulated in ChP (Supplementary Fig. 2b–g). Gene sets associated with translation and active mitochondria were enriched at 9 p.m. (Fig. 2f, g), consistent with the abovementioned translation changes and crosstalk between metabolism and circadian clocks[33,34]. These data raise the hypothesis that ChP metabolism responds to environmental cues and correlates with increased translation activity. We also identified diurnal changes in ChP gene sets associated with barrier adhesion and permeability. These transcripts were largely upregulated at 9 a.m. (Fig. 2h), suggesting that the ChP epithelial barrier changes throughout the day, consistent with diurnal variations in ChP function. Taken together, these results show that the ChP translatome and secretome are regulated diurnally, with differential effects on translational machinery, secreted proteins, metabolism, and barrier components.

### TTR accumulates in the ChP during the dark phase and depends on Bmal1 and feeding cues

Of the top 5 dark phase-secreted proteins identified to be diurnally regulated at the level of translation, the mostly highly expressed gene

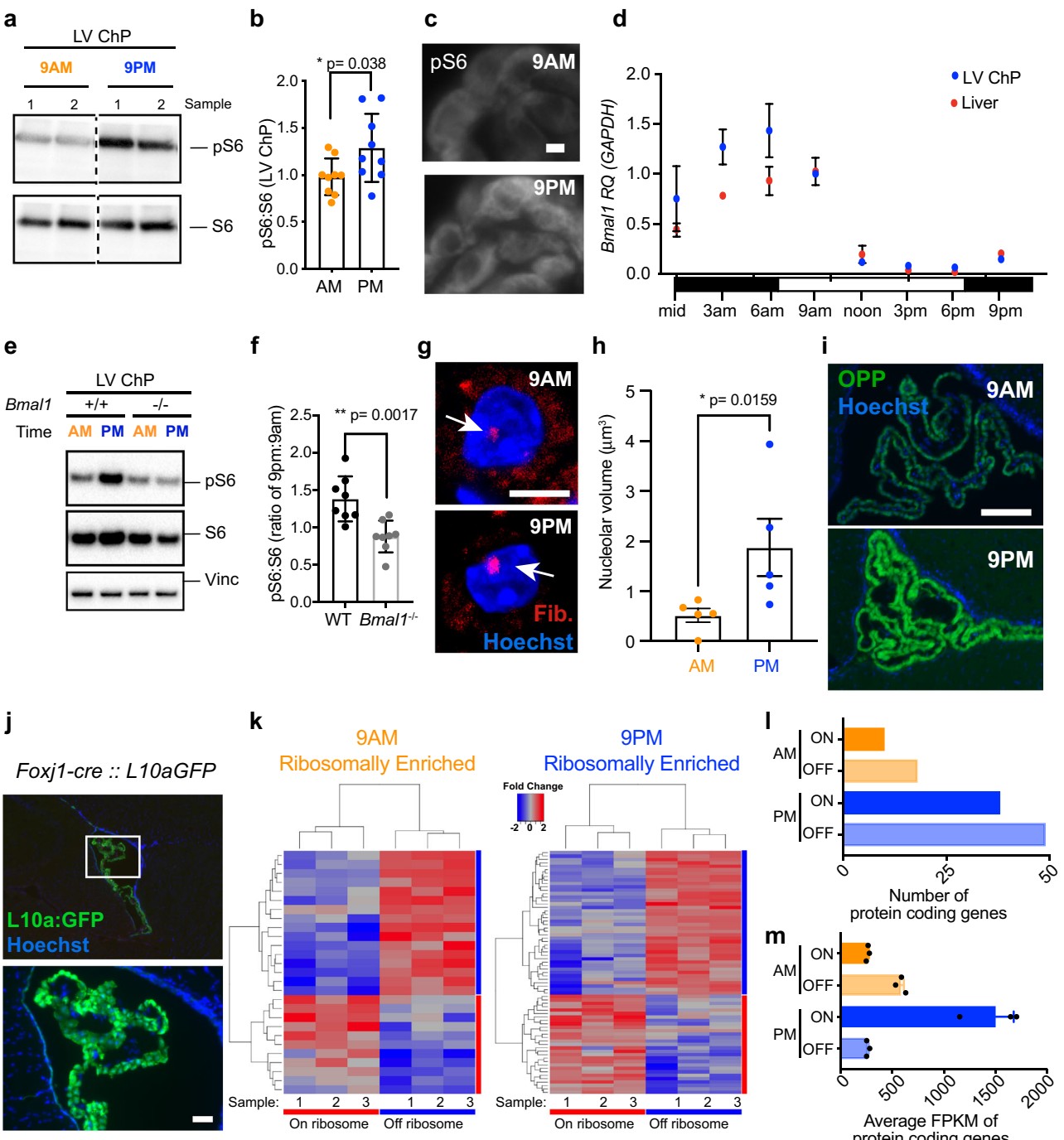

was Transthyretin (*Ttr*) (Fig. 3a). *Ttr* is a signature gene of the ChP that encodes a carrier protein for thyroid hormone, amyloid-β, and retinoic acid[35–37]. We validated the TTR expression pattern by immunoblotting at 9 p.m. vs. 9 a.m. (Fig. 3b, Supplementary Fig. 3c). A recent study in rat ChP identified about 1.5× rhythmic variation of *Ttr* transcript[38], where the nadir and zenith were equivalent to noon and midnight, respectively. However, in our experiments in mouse, the modulation of *Ttr* expression was not reflected at the level of transcript abundance (Supplementary Fig. 3a, b, i). Rather our data indicate that ribosomal association alone is likely to account for diurnal *Ttr* expression modulation (Fig. 3a).

To uncover how ChP-TTR is regulated throughout the day, we first used *Bmal1*-null mice with a disrupted molecular clock. Diurnal ChP-TTR protein expression depended on the molecular clock and was altered in ChP from *Bmal1*-null mice (Fig. 3c). Since the intrinsic cell-

autonomous clock is absent in *Bmal1*-null mice, all circadian behaviors, including feeding, are disrupted[39,40]. Therefore, this model could not discern whether the observed ChP molecular clock was entrained by pacemaker-driven intrinsic cues alone or secondarily to feeding or other nutrition-dependent behaviors. To distinguish between these variables, we took advantage of the fact that feeding cues have the ability to entrain many peripheral clocks[34] and tested the in vivo effects of switching from an ad libitum feeding paradigm to time-restricted feeding (Fig. 3d; Supplementary Fig. 3e–h). This feeding restriction paradigm allowed us to decouple feeding-driven metabolic and behavioral clocks from the light-driven SCN clock[39–41]. We found that diurnal regulation of ChP TTR protein levels was dependent on feeding. The high dark-phase TTR protein expression was maintained in ChP from mice receiving dark-phase restricted feeding but was equalized in ChP from mice restricted to light-phase feeding, while

**Fig. 1 | Choroid plexus translation is diurnally regulated to be higher during the dark phase. a** Immunoblotting of ChP protein extracts showed increased S6 phosphorylation (pS6) relative to total S6 at 9 p.m. (blue) compared to 9 a.m (orange). All vignettes were cropped from the same membrane. **b** Ratio of pS6 to total S6 immunoblotting at 9 a.m. (orange) and 9 p.m. (blue). *$p < 0.05$ ($p = 0.038$) Student's two-tailed unpaired $t$ test, $N = 9$ biologically independent animals at each time over 3 independent experiments. Data are presented as mean values ± standard deviation (SD). **c** Immunostaining for pS6 in lateral ventricle (LV) ChP also shows increased pS6 at 9 p.m.; scale bar = 50 μm. **d** RT-qPCR analysis of *Bmal1* expression in LV ChP (blue) and liver (red) showed cycling of the molecular clock and revealed similar phase between the two tissues. Data are presented as mean values ± standard deviation (SD). **e** Immunoblotting of ChP protein extracts for pS6 showed that increased S6 phosphorylation at 9 p.m. is dependent on *Bmal1*. **f** Ratio of pS6 to total S6 from immunoblots of ChP from *Bmal1*-null vs. WT mice. **$p < 0.01$ ($p = 0.0017$) Student's two-tailed unpaired $t$ test, $N = 8$ biologically independent animals at each time over 4 independent experiments. Data are presented as mean values ± standard deviation (SD). **g** Immunohistochemistry of the nucleolar protein Fibrillarin (red) in ChP epithelial cells showed larger nucleoli at

9 p.m., during the dark phase; scale bar = 5 μm. $N = 4$ at each time. **h** Quantification of median nucleolar volume. *$p < 0.05$ Student's two-tailed unpaired $t$ test, $N = 5$ biologically independent animals at each time over 2 independent experiments. Data are presented as mean values of the median of 50 nucleoli per animal ± standard error of the mean (SEM). **i** OPP incorporation assay in adult ChP epithelial cells showed an increased rate of protein synthesis at 9 p.m. than at 9 a.m.; scale bar = 100 μm. **j** RPL10A-conjugated EGFP expression in ChP epithelial cells after *Foxj1-Cre* recombination in TRAP-BAC mice; scale bar = 100 μm. **k** Heatmaps and hierarchical clustering of transcripts associated with RPL10A vs. those in the supernatant at either 9 a.m. or 9 p.m. (adjusted $p < 0.05$). **l** Number of distinct, unique protein-coding genes on (solid) or off (transparent) the RPL10A ribosomal subunit at 9 a.m. (orange) and 9 p.m. (blue) (adjusted $p < 0.05$). **m** Average FPKM of protein-coding genes on (solid) or off (transparent) the RPL10A ribosomal subunit at 9 a.m. (orange) and 9 p.m. (blue) (adjusted $p < 0.05$). $N = 3$ biologically independent samples (LV ChP pooled from three animals per sample) at each time in 1 experiment. Data are presented as mean values ± standard deviation (SD). Male mice were analyzed. Source data are provided as a Source Data file (Source Data).

---

transcript levels were unaffected (Fig. 3d, Supplementary Fig. 3e–h). These results demonstrate that the ChP responds to external cues (e.g., nutrition or activity) to modulate its own output, in concert with the molecular clock. These results both suggest that the ChP and CSF systems functionally respond to complex nutritional and behavioral input and provide a framework for testing roles of specific inputs into the system.

### Tracking ChP and TTR rhythmicity ex vivo in ChP explants

Because ChP translation changed diurnally, we further refined the timeline of ChP output through the day. We used ChP explants[42] from *Per2:luc* mice in which luciferase bioluminescence luminometry reports the expression of the circadian clock gene, *Period2 (Per2)*[43] (Fig. 3e). We tracked ex vivo ChP rhythmicity over 14 days, and from these readouts, calculated the average period length to be just under 24 h (Fig. 3e–g). This period length was indistinguishable between endogenous explants and after addition of the potent zeitgeber dexamethasone (Fig. 3f, g), which, consistent with its endogenous ability to synchronize clocks[44], resynchronized the tissue without altering the period length. In parallel, we found significant circadian oscillations in key clock components *Per2* and *Bmal1* by RT-qPCR (Supplementary Fig. 1, S3a). These results corroborate previous studies showing that ChP maintains endogenous rhythmicity[14,17], and add the observation that ChP tissue is responsive to extrinsic glucocorticoid cues.

To test for circadian expression of TTR within individual animals rather than from cumulative terminal samples collected at each time, we used genome editing to generate a *Ttr^mNeonGreen* reporter mouse where TTR harbors a C-terminal mNeonGreen tag (Fig. 3h, Supplementary Figs. 3d, 4a). The protein appropriately localized to ChP epithelial cells (Fig. 3h, Supplementary Fig. 3d), and was secreted into CSF (Supplementary Fig. 4b). Ex vivo 3-day imaging of ChP explants generated from *Ttr^mNeonGreen* mice indicated that TTR protein levels in ChP tissue cycle across circadian time (Fig. 3i, j). We validated these findings by immunoblotting lysates of ChP collected at 3 h intervals. TTR protein was substantially upregulated in ChP at 9 p.m., 2 h after lights-off, and remained relatively high throughout the dark phase (Fig. 3k), as in liver (Supplementary Fig. 3j). Using multi-day ex vivo monitoring, we confirmed prior reports of ChP circadian rhythmicity ex vivo (Myung et al., 2018), and then developed an imaging pipeline to track endogenous patterns of ChP-TTR expression in real-time. Collectively, we demonstrate that ChP-TTR diurnal protein level variations are post-transcriptionally regulated and suggest that ChP-TTR expression peaks around lights-off.

We then asked how ChP-TTR affected CSF-TTR and found that after the onset of the dark phase, CSF-TTR levels (relative to the stably expressed iron binding protein Transferrin (Tf)), increased with a slight delay such that by 10 p.m., CSF-TTR was increased compared to

9 p.m., and CSF-TTR remained high at 11 p.m.[45–47] (Fig. 3l). Because a key role of TTR is to transport thyroid hormone, we hypothesized that levels of CSF thyroid hormone would correspond with changing TTR availability. To test this idea, we developed a liquid chromatography-mass spectrometry (LC-MS) method for CSF thyroid hormone detection, where we optimized several parameters, including chromatography-based analyte separation, ionization, detection, and chemical preservation. Focusing on the time during which the TTR transition occurs, between 6 pm and midnight (Fig. 3k), we found that increased CSF-TTR was accompanied by concurrent changes in multiple polar metabolites in CSF including an increase in active thyroid hormone T3 (triiodothyronine) (Fig. 3m) at 9 p.m., but not a significant change in the circulating form T4 (thyroxine) (Supplementary Fig. 3k, l). Collectively, application of several analysis modalities revealed that global changes in ChP protein expression track in register with CSF-TTR levels. Consistent with a role for CSF-TTR in thyroid hormone transport and retention, CSF-T3 increased during the dark phase when CSF-TTR was higher. Because thyroid hormone levels play critical roles in neurodevelopment and metabolism as well as in depression, mania and other psychiatric manifestations[48–52], these data may have implications for thyroid sensitive brain functions and pathology.

### Real-time tracking of CSF-TTR levels throughout the day in awake mice

We sought to determine whether the time course of changes in ChP TTR resulted in parallel changes in TTR secretion into the CSF. Serial CSF sampling for immunoblotting is not suitable for this experiment because as a terminal procedure, it introduces both inter-individual variability and stress-related handling artifacts which may impair measurement accuracy. We overcame these challenges by adapting fiber photometry to the CSF in the *Ttr^mNeonGreen* mouse line. Adult wild-type and *Ttr^mNeonGreen* mice were outfitted with bilateral LV cannulae and optical fibers for freely-moving photometry recordings from CSF lasting between 24 and 96 h (Fig. 4a, b, Supplementary Fig. 4c–f). CSF mNeonGreen fluorescence showed a robust and sustained increase across the first lights-off transition, with the rise in CSF-TTR signal beginning up to two hours before the lights were extinguished (Fig. 4c). More than half of the *Ttr^mNeonGreen* photometry recordings demonstrated peak response magnitudes (defined as the average signal between 8:30 p.m. and 9:30 p.m.) above the range of auto-fluorescence levels observed in wild-type mice. Among these recordings showing robust increases in TTR levels, 20% of the peak signal was achieved a median of 66.9 min before lights-off (Fig. 4d–f, Supplementary Fig. 4g). The real-time readout of CSF fluorescence enabled by the photometry method thus demonstrates that the diurnal rhythm of CSF-TTR anticipates the light transition rather than being a direct

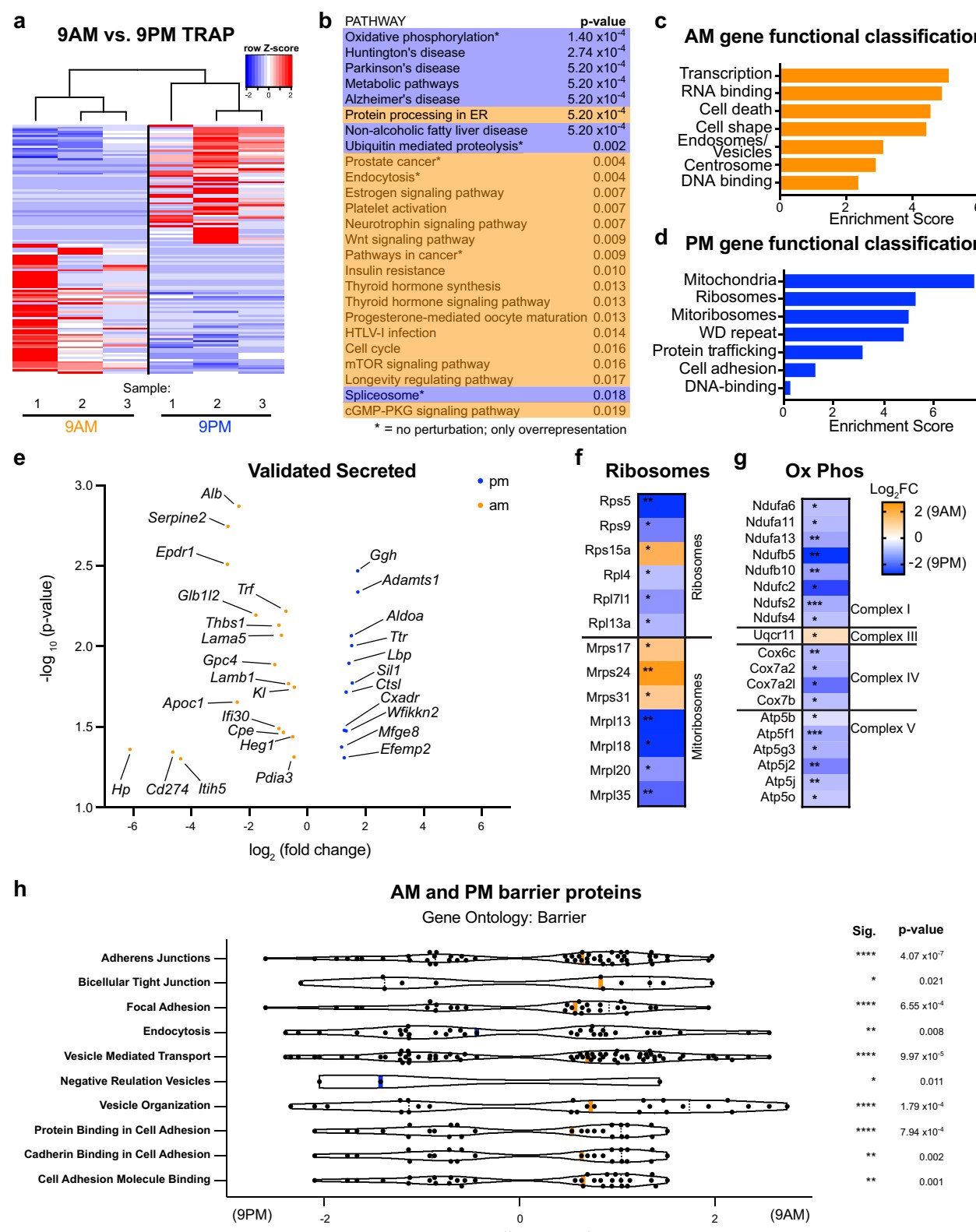

**Fig. 2 | Association of ChP cytoplasmic, membrane bound, and secreted protein mRNAs with the ribosome is diurnally regulated. a** Heatmaps and hierarchical clustering of z-scores for transcripts associated with RPL10A in LV ChP at 9 a.m. vs. 9 p.m. (adjusted $p < 0.05$). CuffDiff; adjusted $p < 0.05$; |log2 FC|>0.4. **b** All significantly regulated pathways from pathway enrichment and overrepresentation analysis (corrected for false discovery rate (FDR). **c, d** Top 7 enriched functional annotation clusters by DAVID in LV ChP at 9 a.m. (orange) and 9 p.m. (blue). **e** Volcano plot of significantly enriched RPL10A-associated transcripts with a product predicted to be secreted from LV ChP at 9 a.m. (orange) and 9 p.m. (blue).

$*p \leq 0.01$; $**p \leq 0.001$; $***p \leq 0.0001$; $****p \leq 0.00001$; adjusted $p$ values. **f** TRAP data for the individual genes associated with ribosomes and mitoribosomes. $*p \leq 0.01$; $**p \leq 0.001$; adjusted $p$ values. **g** TRAP data for the individual genes associated with oxidative phosphorylation. $*p \leq 0.01$; $**p \leq 0.001$; $***p \leq 0.0001$; $****p \leq 0.00001$; adjusted $p$ values. **h** TRAP data for the enriched pathways associated with barrier permeability. The median log$_2$ fold change value is indicated by the solid vertical bar. $*p \leq 0.01$; $**p \leq 0.001$; $***p \leq 0.0001$; $****p \leq 0.00001$; adjusted $p$ values. Male mice were analyzed. Source data are provided as a Source Data file (Source Data).

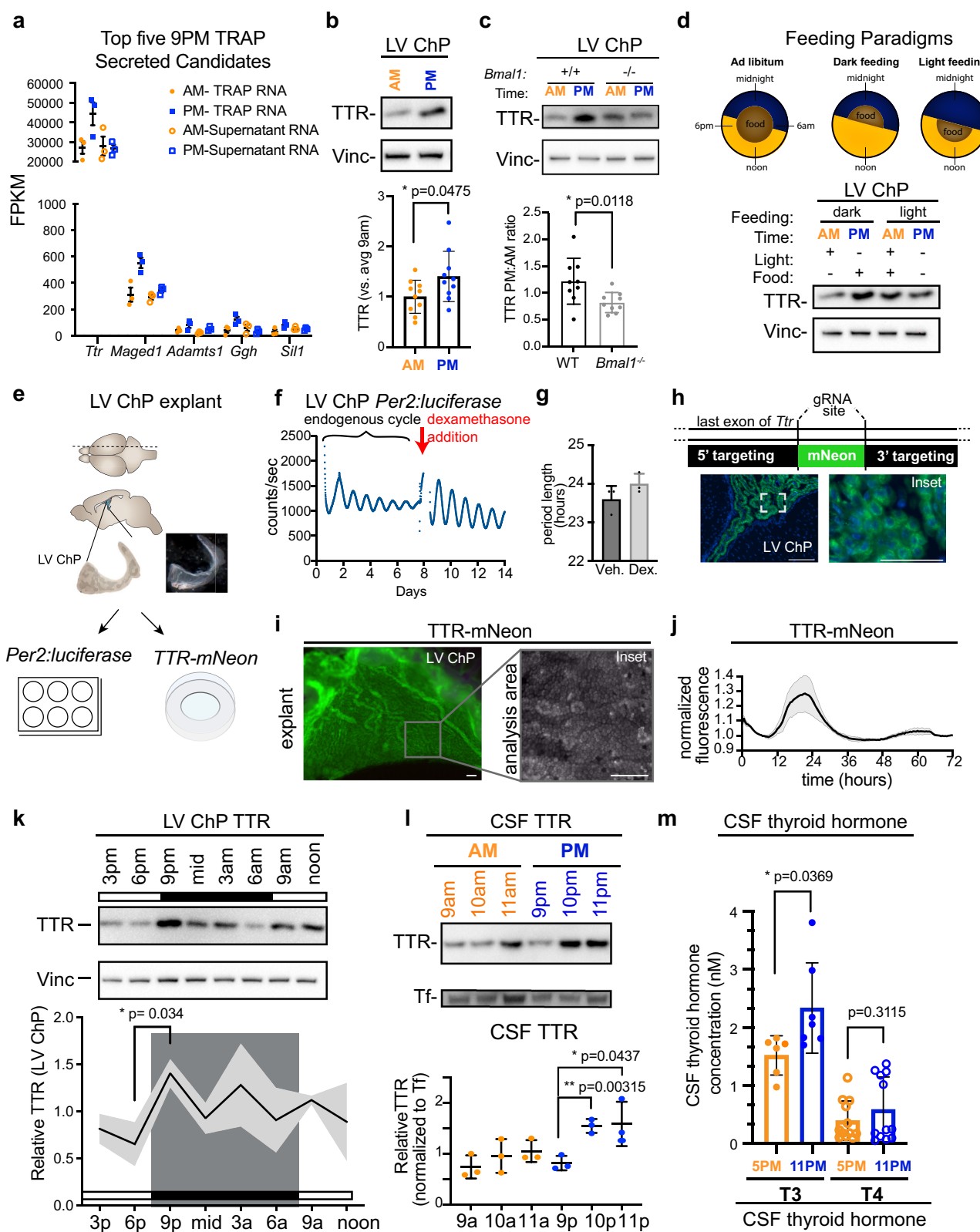

consequence of light-sensitive brain activity. This result suggests that CSF-TTR is likely regulated by an intrinsic timing mechanism. Notably, a smaller subset of mice demonstrated a nearly flat signal within the range of wild-type variability, and several outliers showed a slow decrease in mNeonGreen signal over the same timescale, likely reflecting inter-individual differences in cannula placement as well as diurnal rhythm disruptions due to handling stress.

Next, we continuously monitored mNeonGreen fluorescence across several days. We found that the trend of increasing CSF fluorescence in the night and decreasing fluorescence in the day persisted both during normal light–dark cycling as well as during complete darkness with a period just under 24 h (Fig. 4b, Supplementary Fig. 4h–j). Notably, despite a decay of the mNeonGreen signal over tens of hours due to photobleaching, the modulation amplitude over a

**Fig. 3 | *Ttr* is preferentially translated by the ChP during the dark phase and is dependent on feeding. a** The top 5 TRAP candidates enriched at 9 p.m. from ChP TRAP pulldown. FPKM for each individual mouse is shown at 9 a.m. (orange circles) and 9 p.m. (blue squares) for those transcripts associated with RPL10A (solid circles) and those in the supernatant (empty circles). These are candidates for preferential regulation at the level of translation. $N = 3$ biologically independent samples (LV ChP pooled from 3 animals per sample). Data are presented as mean values ± SEM. **b** Immunoblotting analysis and quantification of LV ChP expression of TTR protein. *$p < 0.05$ ($p = 0.0475$); Student's two-tailed unpaired $t$ test, $N = $ LV ChP from 10 biologically independent animals at each time over 3 independent experiments. Data are presented as mean values ± standard deviation (SD). **c** Immunoblotting of ChP protein extracts for TTR showed that increased protein levels at 9 p.m. is dependent on *Bmal1*. Quantification of the TTR intensity ratio between 9 a.m. and 9 p.m. in WT and *Bmal1$^{-/-}$* animals. *$p < 0.05$ ($p = 0.0118$); Student's two-tailed unpaired $t$ test, $N = 8$ ratios from biologically independent pairs of animals from each genotype over 3 independent experiments. Data are presented as mean values ± standard deviation (SD). **d** Schematics of the restricted feeding regimes used as related to the *ad libitum* paradigm that was used for the previous studies up until this point. Immunoblotting of ChP protein extracts for TTR showed that increased protein levels at 9 p.m. is dependent on feeding behavior. **e** Experimental setup for explant studies including PER2:LUC luminometry and TTR:mNeonGreen microscopy. **f** Representative PER2:LUC oscillations in isolated culture before and after addition of dexamethasone. **g** Average period length for PER2:LUC ChP oscillations in isolated culture before and after dexamethasone are

close to 24 h. $N = 3$ ChP over 2 independent experiments. Data are presented as mean values ± SEM. **h** Genetic targeting used to generate *Ttr$^{mNeonGreen}$* mouse showing appropriate distribution of mNeonGreen to ChP epithelial cells. Scale bar = 100 μm; inset scale bar = 50 μm. **i** Explant preparation from *Ttr$^{mNeonGreen}$* ChP (scale bar = 100 μm; inset scale bar = 100 μm) shows **j** rhythmic oscillations of mNeonGreen across the day ex vivo. $N = 2$ LV ChP over 2 independent experiments. Data are presented as mean values normalized to the final value and shaded area represents range. **k** Immunoblotting of TTR in LV ChP across a whole day at 3-h intervals shows sharp upregulation of TTR in ChP during the dark phase. *$p = 0.034$ corrected for 8 comparisons with Šídák's multiple comparisons test. Solid line represents average (normalized to vinculin and TTR average value) and shaded area represents standard deviation (SD). $N = 3$ biologically independent animals at each time across 3 independent experiments. **l** Immunoblotting of constant volumes of CSF shows sharp upregulation of TTR in CSF at 10 p.m. and 11 p.m. relative to transferrin (Tf). *$p < 0.05$ ($p = 0.0437$), *$p < 0.01$ ($p = 0.00315$); Student's two-tailed unpaired $t$ test. $N = 3$ biologically independent animals at each time across 2 independent experiments. Data are presented as mean values ± standard deviation (SD). **m** CSF concentration of active thyroid hormone T3 (triiodothyronine) and the circulating form T4 (thyroxine) at 5 p.m. (orange) and 11 p.m. (blue). *$p < 0.05$ ($p = 0.0369$), Student's two-tailed unpaired $t$ test. $N = 7$ biologically independent animals at each time in one experiment. Data are presented as mean values ± standard deviation (SD). Male mice were analyzed. Source data are provided as a Source Data file (Source Data).

24-h cycle remained significantly greater in *Ttr$^{mNeonGreen}$* mice relative to wild-type controls a day or more into the recording, indicating that the signal remained above background autofluorescence (Supplementary Fig. 4k). Collectively, these results suggest that the diurnal rhythm of CSF TTR is not strictly determined by the light cycle, but the pattern of expression is maintained even in the free-running state, when the light cues disappear, suggesting a level of autonomy from the light-driven clock cues.

## ChP metabolism is diurnally regulated

In addition to secreted proteins, mRNA transcripts encoding ribosome components, mitoribosome subunits, and components of the electron transport chain oxidative phosphorylation pathway were preferentially associated with ribosomes during the dark phase (Figs. 2b, f, 5a–c). Since many components involved in aerobic respiration were upregulated at 9 p.m., we hypothesized that ChP capacity for oxygen consumption also differed across times of day. We used Agilent Seahorse XFe technology to monitor oxygen consumption as an index of the metabolic status using LV ChP explants taken from mice at 9 a.m. and 9 p.m. (Supplementary Fig. 5c–e). While we identified large changes in metabolic components from serial samples of acutely collected ChP (Fig. 5b, h, i), we did not observe functional metabolic differences in oxygen consumption or ATP production between 9 a.m. and 9 p.m. in ChP explants in constant 0.18% glucose (Supplementary Fig. 5d, e). For reference, normal mouse blood glucose is 80–100 mg/dL (0.08–0.1%) between fasting and feeding, and normal CSF glucose is usually ~60% of the plasma level[53]. We presume that explanted ChP tissue rapidly adapted to the glucose-rich media of the assay, precluding accurate testing of diurnal ChP respiration ex vivo.

We overcame these ex vivo technical hurdles by adapting LC-MS targeted metabolomics that interrogated 250 small molecules covering major central carbon metabolism pathways, which allowed us to investigate in vivo changes in ChP metabolism. ChP tissues were collected at 9 a.m. and 9 p.m. and immediately processed and analyzed on our metabolomics platform, which allowed for a more precise and unbiased view of the tissue's metabolome. Independent clustering reproducibly segregated the two populations of tissue and identified significantly different metabolites at these two times (Fig. 5d, Supplementary Fig. 5f, g). Consistent with differentially expressed citric acid cycle (TCA) genes (Fig. 5b), metabolites of the TCA reflecting oxidative phosphorylation through the electron

transport chain (ETC) were increased during the dark phase (Fig. 5e). By contrast, metabolites indicating shuttling of glucose to the pentose phosphate pathway (PPP) accumulated in ChP during the light phase (Fig. 5f). This major metabolic shift indicates that local ChP metabolism is an output of diurnal cues (intrinsic or extrinsic). Strikingly, redox metabolites in the ChP, including NADH, NADPH, and GSH, were preferentially detected in their oxidized state during the dark phase, with accumulation of products of the ETC including ATP (Fig. 5g, Supplementary Fig. 5h) and a consistent decrease in the reduced:oxidized ratio of major redox metabolites including NADPH:NADP$^+$, GSH:GSSG and NADH:NAD$^+$ (Fig. 5g). The increased ATP and oxidated species are consistent with the increased mTOR-effector signaling observed during the dark phase as mTORC activation mediates oxidative phosphorylative metabolism[54] (Fig. 1a). In addition to reflecting diurnal oxidative shifts, this newly generated differential ChP metabolite dataset is hypothesis-generating. For example, significant diurnal shifts in 2-hydroxyglutarate levels could suggest circadian differences in important physiological processes including responses to hypoxia and chromatin modifications[55] (Supplementary Fig. 5i).

Consistent with higher levels of oxidative metabolism during the dark phase and with ChP-TTR levels depending on metabolic cues like feeding, the citric acid cycle enzyme Citrate synthase (CS), oxidative phosphorylation (OXPHOS) components (CV-ATP5A; CIII-UQCCRC2; CIV-MTCO1; CII-SDHB; C1-NDUFB8), and Mitofusin2 (MFN2) were all upregulated at 9 p.m. in LV ChP (Fig. 5h). In addition, the amount of CS within individual ChP epithelial cells was increased (Fig. 5i), although the number of mitochondria remained unchanged in each ChP epithelial cell (Supplementary Fig. 5a, b). Collectively, our data suggest that the ChP maintains an endogenous circadian rhythm that 1) diurnally regulates metabolic processes including TCA, ETC, and overall oxidative state, and 2) adapts on a relatively fast timescale in response to external systemic nutritional cues.

## ChP barrier components and microstructure are diurnally regulated

ChP epithelial cells comprise the blood-CSF barrier, a key brain barrier regulating selective access of systemic cues to the CNS[56]. Because barrier components were differentially regulated in our diurnal TRAP data, we next tested whether the blood-CSF barrier properties are diurnally regulated. Extracellular matrix (ECM) and adhesion proteins

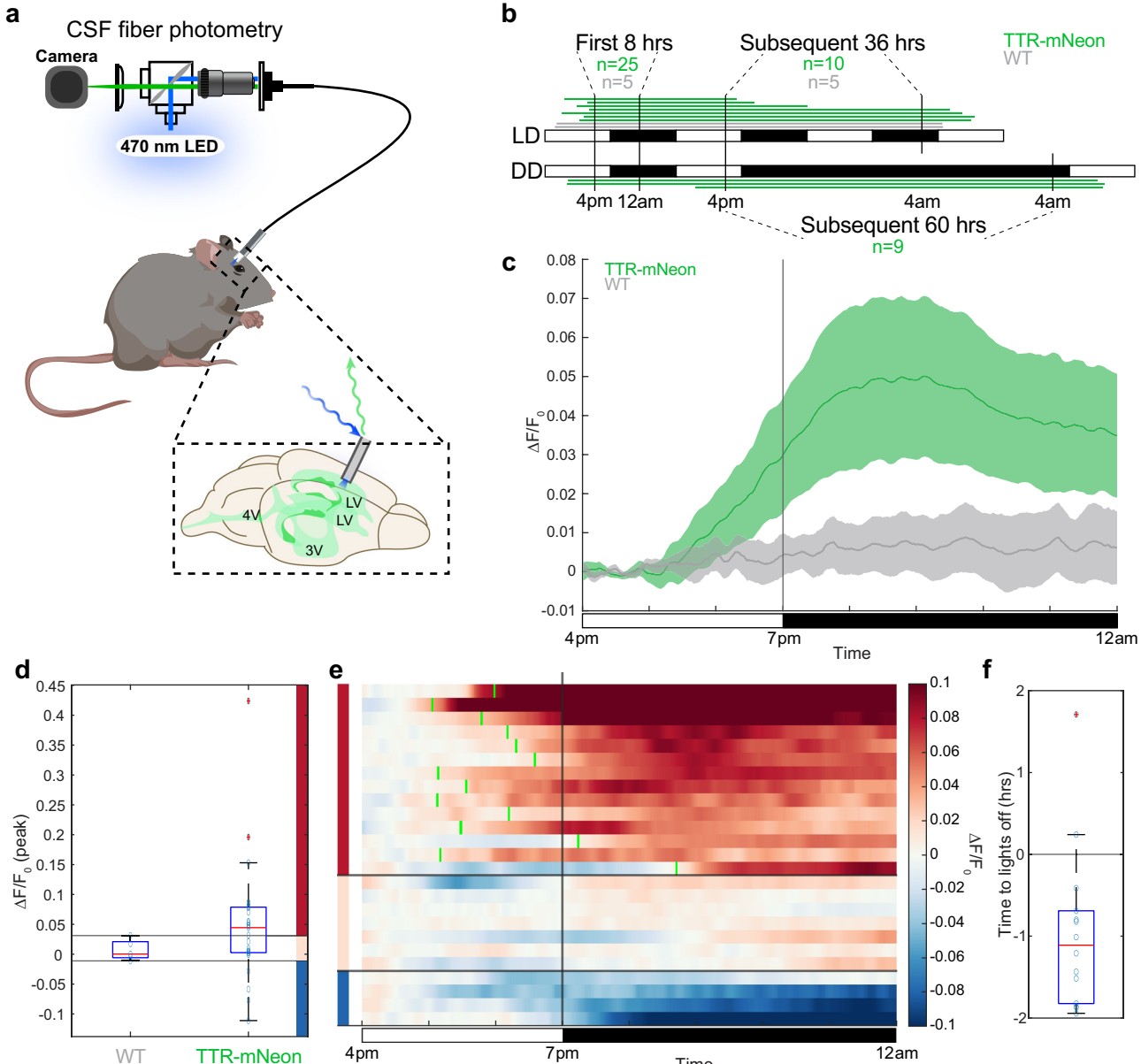

**Fig. 4 | CSF TTR is dynamic across the day. a** Schematic of the freely-moving photometry setup, consisting of a 470 nm LED light source, CMOS camera for signal collection, and patch cord to the animal. Inset shows the geometry of the optical fiber in the cannula relative to the ventricular system of the mouse. **b** Overview of all summarized photometry recordings, with each bar indicating the start (on the left) and stop times (three animals recorded simultaneously in each recording). The first 8 h (light-dark transition) of both the light–dark and dark-dark recordings were combined for analyses and subsequently treated separately for the later periods when the light patterns were distinct. Two $Ttr^{mNeonGreen}$ and one wild-type animal were excluded from analysis for health reasons. **c** Summary of all $Ttr^{mNeonGreen}$ and wild-type recordings in the first 8-h period around the first lights-off. $N = 25$ biologically independent $Ttr^{mNeonGreen}$ animals across seven independent experiments and $N = 5$ biologically independent wild-type animals across two experiments. Data are presented as mean values ± standard error of the mean (SEM). **d** Peak responses, defined as the average signal over the 8:30 p.m.−9:30 p.m. interval, of all $Ttr^{mNeonGreen}$

and wild-type recordings summarized in **c**. Horizontal lines are drawn at the minimal and maximal wild-type signals, and $Ttr^{mNeonGreen}$ recordings are categorized based on these subdivisions into strong responders (top group, red), wild-type-like responders (middle group, peach), and dropping signals (bottom group, blue). **e** Heatmap showing each individual recording summarized in **c**, with subgroups color-coded as in **d**. Green tick marks on the heatmap indicate the times where each strongly responding trace reaches 20% of its peak signal (average between 8:30 p.m. and 9:30 p.m.). **f** Summary of the green tick marks in the heatmap in **e**, indicating that 20% peak response is reached a median of 66.9 min before lights-off. Box plots in **d**, **f** show median value at the red central bar, with the bottom and top edges of the box indicating the 25th and 75th percentiles, respectively. The whiskers extend to the most extreme values not considered outliers, and outliers (more than 1.5 times the interquartile range away from median) are plotted individually and marked with a red '+' symbol. Male mice were analyzed. Source data are provided as a Source Data file (Source Data).

are necessary for appropriate circadian rhythmicity in other epithelia[57–60] and in the BBB[61]. Pathway enrichment analysis (Advaita) of our data revealed endocytosis ($p = 0.004$, FDR correction), and gene ontology (GO) enrichment analysis identified vesicle-mediated transport ($p = 9.97 \times 10^{-5}$, FDR correction) to be significantly upregulated in ChP during the day (Figs. 2h, 6a, S6a), consistent with a more

permeable blood-CSF barrier during the light phase. Barrier components showed TRAP profiles consistent with higher levels of translation during the light phase (Fig. 6a). While four of these candidates demonstrated mild 24 h rhythmicity at the transcript level (Fig. 6b) [integrin beta-8 (*Itgb8*), solute carrier family 7 member 8 (*Slc7a8*), cadherin3/p-Cadherin (*Cdh3*), ATP binding cassette subfamily F

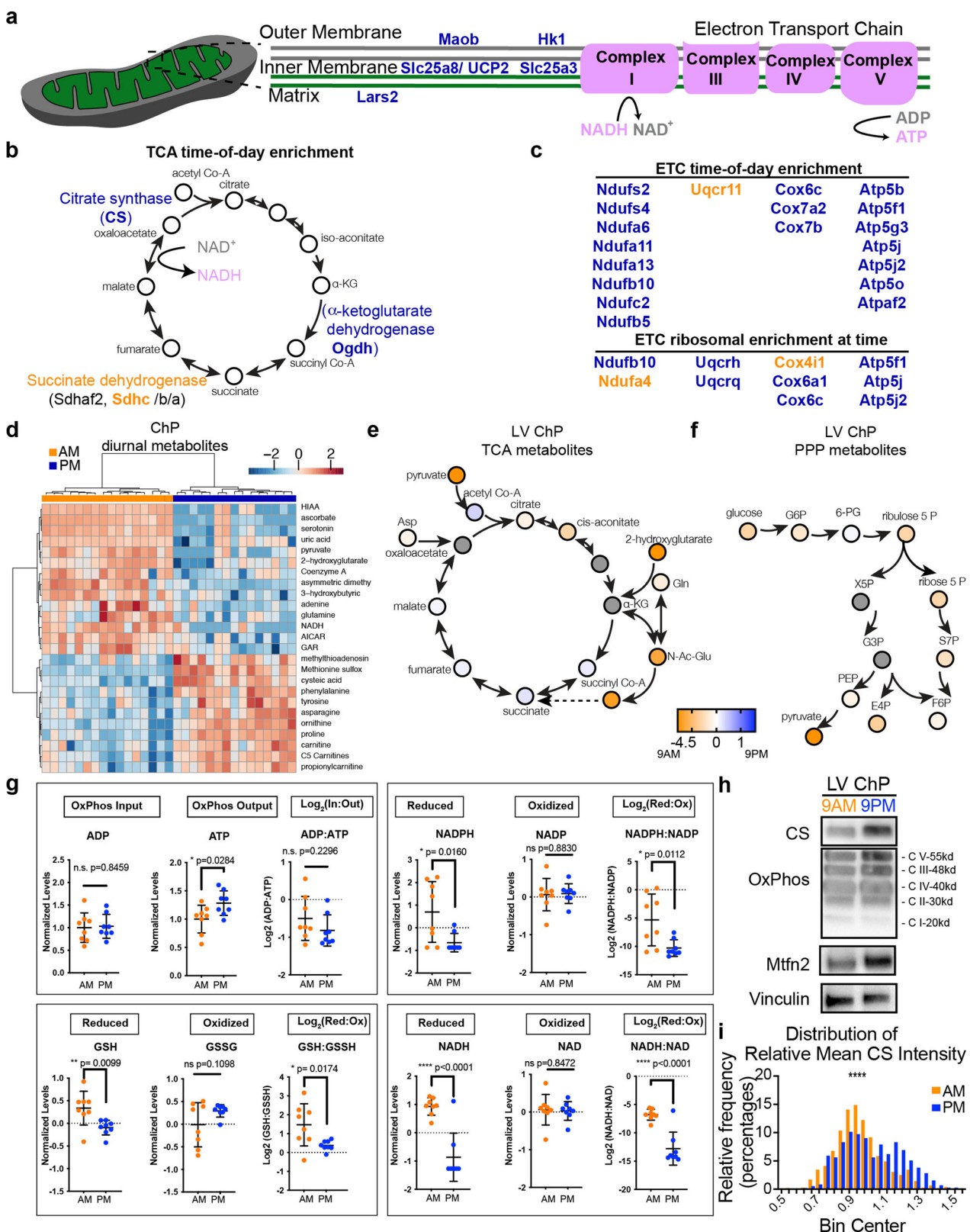

member 3 (*Abcf3*), and intracellular trafficking component sorting nextin 12 (*Snx12*)], transcriptional regulation alone is insufficient to explain the drastic variation in their ribosomal loading between 9 a.m. and 9 p.m. In fact, *Cdh3* was the only one of these genes to have a peak in transcript level in the middle of the day. The other genes showed a striking phase reversal in that transcript levels peaked around 9 p.m. while ribosomal loading was clearly higher at 9 a.m. Notably, *Chmp1b*

mRNA was not rhythmic even though TRAP revealed differential ribosome association of its transcripts. These data suggest that translational circadian regulation can be significantly out of phase from transcript rhythmicity, an important consideration in circadian studies relying on transcriptional readouts (Fig. 6b, Supplementary Fig. 6a). Diurnally regulated solute carriers included neutral amino acid transporter *Slc7a8* (Lat2) known to maintain minimal CSF amino acid

**Fig. 5 | ChP metabolic components and metabolites are diurnally regulated.**
**a**–**c** Schematics of the mitochondrial transport, the citric acid (TCA) cycle, and electron transport chain components. Those components that show altered enrichment on the ribosome at either 9 a.m. (orange) or 9 p.m. (blue). Listed components include those that are significantly associated with ribosomes at either 9 a.m. or 9 p.m. and those that are enriched on ribosomes vs. off ribosomes at either 9 a.m. or 9 p.m. **d** Heatmap of top 25 changed metabolites in 9 a.m. vs 9 p.m. ChP. **e** Metabolite $\log_2$ (ratio) for TCA intermediates in 9 a.m. and 9 p.m. LV ChP. $N = 8$ biologically independent animals at each time in one experiment. **f** Metabolite $\log_2$ (ratio) for pentose phosphate pathway (PPP) intermediates in 9 a.m. (orange) and 9 p.m. (blue) LV ChP. $N = 8$ biologically independent animals at each time in one experiment. **g** Values of common redox electron donor and recipient pairs followed by the $\log_2$ (reduced: oxidized ratio) showing increased oxidation at 9 p.m. (blue) in LV ChP. $N = 8$ biologically independent animals at each time in one experiment; Student's two-tailed unpaired $t$ test. Data are presented as mean values ± standard deviation (SD). **h** Immunoblotting for citrate synthase (CS), core components of the electron transport chain (OXPHOS; CV-Atp5a; CIII-Uqccrc2; CIV-Mtco1; CII-Sdhb; C1-Ndufb8), mitofusin2 (Mtfn2) showed enriched mitochondrial components during the 9 p.m. dark phase. **i** Quantification of citrate synthase (CS) intensity in immunohistochemistry of LV ChP epithelial cells showed a shift toward higher intensity expression of CS at 9 p.m. (blue) than at 9 a.m. (orange). ****$p < 0.0001$; Kolmogorov–Smirnov test, $N = 5$ biologically independent animals at each time across 2 independent experiments. Male mice were analyzed. Source data are provided as a Source Data file (Source Data).

concentrations[62] and zinc transporters (*Slc39a9, Slc30a5, Slc30a1*) necessary for function of other epithelia[63] (Fig. 6a, b, S6a). These data support a potential permeability, absorption, or clearance role associated with the late afternoon peak of these family members in ChP (Fig. 6a, b).

To test this idea, we next compared ChP epithelial barrier properties at 9 a.m. vs 9 p.m. by adapting a traditional blood–brain barrier assessment technique[64] of intravascular horseradish peroxidase (HRP) labeling to observe ChP transcytosis and tight junction morphology (Fig. 6c). We did not observe significant differences in the number of transcytosed vesicles from the blood into ChP epithelial cells along the basolateral surface of the cell-cell junction (Fig. 6d, e, Supplementary Fig. 6b). While we noted a trend toward more vesicles localized near the apical surface during the light phase, there was variability among individuals (Fig. 6f, Supplementary Fig. 6c). The apical tight junction, however, was significantly and substantially wider during the 9 a.m. light phase (Fig. 6g, h, Supplementary Fig. 6d, e), potentially indicating a more permeable ChP barrier during the light phase. Taken together, our results show rhythmicity in structural components that support a model that the blood-CSF barrier, like the blood–brain barrier (BBB), exhibits higher permeability during the light phase[65]. This model suggests potential implications for differential CSF/ brain access of drugs and systemic cues throughout the day. This sensitive set of analyses can be broadly applied to query ChP barrier permeability changes in other states and circumstances.

## Discussion

We developed, curated, and adapted a set of multi-modal approaches that comprise an integrated toolkit for analyzing the ChP-CSF axis across circadian time and in response to associated environmental cues. Additionally, data generated from low-abundance samples of mouse ChP and CSF analyzed with this toolkit represent valuable benchmarks for how the ChP and CSF change throughout the day and respond to external cues. We used TRAP ribosomal pulldown, ex vivo ChP explant luminometry and microscopy, in vivo CSF fiber photometry using a $Ttr^{mNeonGreen}$ reporter mouse generated for this purpose, CSF and ChP metabolomics, and ChP barrier analysis. Using these tools, we demonstrated that ChP translation and protein synthesis were regulated diurnally, with higher levels of translation taking place during the dark phases when these animals are more active. This change in translation involved distinct sets of proteins synthesized by the ChP during the dark vs. light phases. These proteins were involved in the core ChP functions of secretion, metabolism, and brain barrier properties. We focused on TTR because it emerged as a top candidate from TRAP analysis and is a key component of ChP and CSF. We found that TTR protein abundance responded to feeding cues and required the molecular clock component *Bmal1*. Ultimately, diurnal regulation of ChP-TTR expression modulated CSF-TTR levels. Concurrently, ChP metabolism was substantially different between day and night with more oxidative phosphorylation during the dark phase. The data were consistent with a more permeable ChP barrier during the light phase. Together, these tools described concerted daily changes in the ChP

and CSF with far-reaching consequences for understanding the composition of this essential and surgically accessible fluid.

Protein translation is emerging as a fundamental process by which circadian rhythmicity controls tissue functions throughout the body. For example, mTOR/4E-BP1-mediated translational control regulates entrainment and synchrony of the SCN master clock[66,67]. In liver, ribosome biogenesis, translation, and protein synthesis are downstream of core circadian components responsive to metabolic cues[24,26]. At the cellular level, BMAL1 rhythmically interacts with translational machinery to promote protein synthesis in response to mTOR signaling, thereby directly connecting circadian timing to the control of protein production[25]. Previous efforts to identify downstream effectors of this circadian rhythmicity on ChP output were somewhat inconclusive—mRNA quantification has either identified no change, as in the case of the water channel *Aqp1*[17], or smaller changes for *ApoJ* and *Ttr* in rats[38]. TTR itself could be induced in cultured ChP cells by glucocorticoids, and upregulated in liver, ChP, and CSF by acute and sometimes chronic stress[68]. The bursts in ChP translation that we observed in the dark phase are likely followed by protein clearance (degradation or removal via CSF outflow) to manifest the quick shifts in protein availability. Consistent with this hypothesis, pathway analysis suggests increased ChP ubiquitin mediated proteolysis during the light phase ($p = 0.002$, FDR correction) and previous studies indicate that CSF production and distribution shift diurnally[2,5]. Our finding that ChP diurnal output is regulated post-transcriptionally suggests that nutrient sensing (e.g., via mTOR) in the ChP may be an upstream regulator of ChP diurnal functions including CSF secretion and brain barrier permeability.

The ChP supplies most CSF-TTR[37] and altered CSF-TTR has been implicated in neurologic diseases. For example, low CSF-TTR is associated with depression[52,69,70]. CSF-TTR may be protective against Aβ toxicity as a result of its ability to bind Aβ[71]. Indeed, Aβ clearance changes during sleep vs. waking states[3]. TTR is associated with oxidative stress in other systems[72], potentially linking the metabolic oxidative shifts with dark phase TTR availability. TTR is also the major transporter of thyroid hormone from the ChP epithelial cells into the CSF[36,37,73,74] and can prevent loss of thyroid hormone from the CSF into the bloodstream in hypothyroid animals[75]. Thyroid hormone levels cycle in a circadian manner in serum[76,77] and in brain tissue[78], but previously published data are not conclusive on the origin of thyroid hormone cycling in the brain[51]. Our data indicate that varying ChP-TTR levels modulate thyroid hormone (T3) availability in CSF. The rise in CSF T3 corresponds with higher serum T4, suggesting that dark phase ChP TTR could be upregulated to transport systemic thyroid hormone into the CSF. Further supporting a role for TTR in thyroid hormone availability, pathway enrichment analysis (Advaita) of 9 a.m. vs. 9 p.m. ChP TRAP data identified thyroid hormone synthesis and signaling ($p = 0.013$, FDR correction) as differentially regulated pathways between light and dark phases.

The ChP displayed a concerted diurnal metabolic shift, including increased oxidation signal during the dark phase. Significantly different metabolite profiles were associated with both ChP and CSF

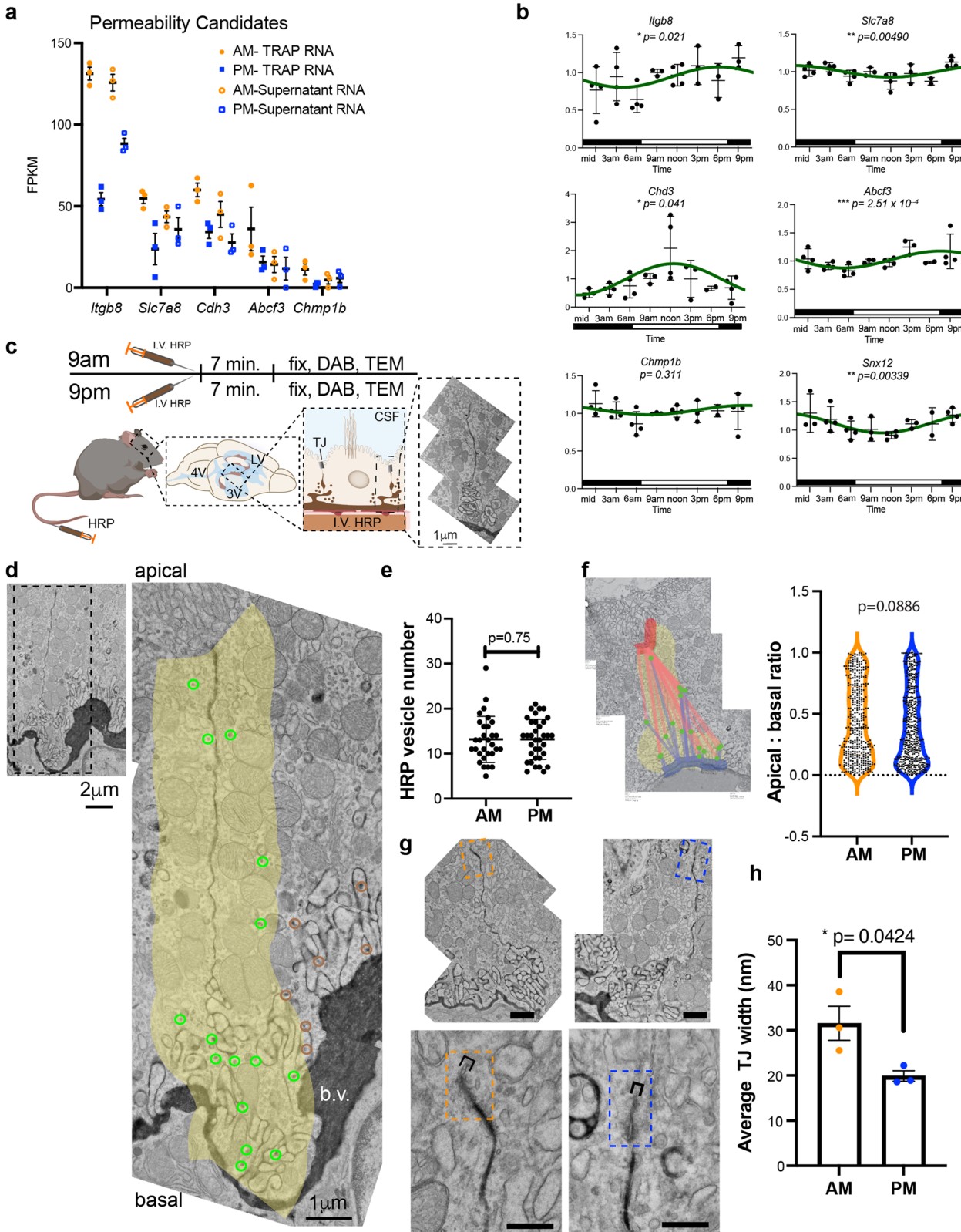

between 9 a.m. and 9 p.m., with others likely to have nadirs and zeniths in accumulation that were not captured by these two timepoints. However, in explants (Supplementary Fig. 5), we presume that the Seahorse media, which contains high glucose, overrode the intrinsic metabolic activity of ChP tissues, especially since we found that at least some ChP circadian properties are feeding dependent (Fig. 3d). This suggests that care should be exercised in explant preparations and is

why subsequent ex vivo analyses were performed on purposefully synchronized tissue and monitored for multiple days. Metabolic responses to diurnal cues are common in other tissues as well, with the circadian clock controlling metabolism at the level of transcription and translation[20,79]. In liver, protein expression of Krebs cycle and oxidative respiratory chain enzymes oscillates, and this rhythmicity is blunted in *Per1/2*-null and *Bmal1*-null mice[33,80]. Metabolic output is also

**Fig. 6 | ChP barrier components and permeability are diurnally regulated.**
**a** TRAP candidates associated with barrier function. FPKM for each individual mouse is shown at 9 a.m. (orange circles) and 9 p.m. (blue squares) for those transcripts associated with RPL10A (solid) and those in the supernatant (open). $N = 3$ biologically independent samples (LV ChP pooled from 3 animals per sample). Data are presented as mean values ± standard deviation (SD). **b** RT-qPCR analysis of the expression of barrier components *Itgb8, Slca8, Chd3, Abcf3, Chmp1b,* and *Snx12* in LV ChP every 3 h. *Itgb8, Slca8, Chd3, Abcf3,* and *Snx12* showed significant rhythmicity by RAIN analysis. $N = 4$ biologically independent animals at each time across 2 independent experiments. **c** Experimental design for barrier permeability assay using intravenous (i.v) injection of horseradish peroxidase (HRP) followed by transmission electron microscopy. **d** Example of HRP-filled vesicles (green circles) taken up during the 7-min incubation period within the peri-junctional area (yellow shading). Vesicles outside of the quantified region are circled in brown.
**e** Quantification of total number of HRP-filled vesicles in the peri-junctional area.

Each point represents one junction, data combined from three animals at each timepoint. Welch's two-tailed unpaired *t* test, $N = 3$ biologically independent animals per time in one experiment. **f** Example of the apical-basal distances calculated for HRP-filled vesicles in the peri-junctional area. Quantification of the (apical-basal)/apical ratio (0 = basal (blue); 1 = apical (red) for HRP-filled vesicles in the peri-junctional area at 9 a.m. (orange) and 9 p.m. (blue). Kolmogorov–Smirnov test, $N = 3$ biologically independent animals per time in one experiment. **g** Example of tight junctions at apico-lateral surface of ChP epithelial cells. Dotted box indicates junction area and bracket indicates width. Scale bars: top = 1 μm; bottom zoom = 500 nm. **h** Quantification of distances between each cell membrane within the tight junction. Each point is the average of 5 distances per junction for 10 junctions in a single animal. *$p < 0.05$; Welch's two-tailed unpaired *t* test, $N = 3$ biologically independent animals per time in one experiment. Data are presented as mean values ± standard error of the mean (SEM). Male mice were analyzed. Source data are provided as a Source Data file (Source Data).

dependent on circadian rhythm in other cell types including cardiomyocytes[81], skeletal muscle[82], hippocampus[83], pancreatic β-cells[84], and macrophages[85]. Gut epithelial metabolism demonstrates circadian rhythmicity, particularly of cytochrome P450 family members[86]. Consistently, our data show higher dark phase cytochrome P450 monooxygenases that target fatty acids/xenobiotics (*Cyp2j6*: $\log_2 FC = -3.047$, $p$ adj = 0.002; *Cyp2u1*: $\log_2 FC = -1.851$, $p$ adj = 0.009). Circadian disruptions link metabolic disease and cognitive decline[87–89]. Our data showing diurnal changes in ChP metabolic components motivate actively integrating ChP into models of these and other diseases. These data corroborate a large body of work indicating metabolic changes across time of day in multiple tissues[34] and place the ChP into the context of systemic metabolism, potentially reflecting its role as a key interface between systemic circulation and the CSF.

The diurnal cycles in cell adhesion components, solute transporters, efflux pumps, and endocytosis that we observed likely modulate barrier properties and could have important clinical implications for chronopharmacology—drug treatment that takes the body's circadian rhythm into consideration. In fact, 85% of trials for drugs with half-lives under 15 h showed dosing time dependence compared with just 39% for longer-acting drugs[90,91]. Some diseases, including psychiatric symptoms, manifest morbidities at specific circadian times[91]. Consistent with our observed ChP permeability changes, BBB permeability including carrier-mediated transport, cell-cell junction permeability, and efflux pumps also varies diurnally, with a more permeable barrier during light phases[56,65]. Recently, neuronal activity (higher during waking hours) was shown to activate endothelial cell ABC efflux transporter expression at the BBB through a *Bmal1*-dependent mechanism[61]. Similarly, intestinal barrier properties change in response to feeding and other circadian cues as is seen in expression of tight junction proteins Occludin and Claudin-1[92]. Because the ChP is a key brain barrier that can modulate drug efflux[56], our data showing diurnal differences in ChP barrier properties have implications for waste clearance, immune cell trafficking, and CNS drug efflux/ influx.

Abnormal CSF components have been associated with a number of neurologic conditions that co-present with circadian disruptions, including hydrocephalus, autism spectrum disorder, Alzheimer's disease, bipolar disorder, and schizophrenia[93–99]. For example, sleep disruption is a diagnostic criterion for major depression, bipolar disorder, post-traumatic stress disorder, generalized anxiety, and other mood disorders[100]. While CSF components, including those that are diurnally regulated, can originate from outside the ChP (e.g., peripheral adrenal glands (cortisol), hypothalamus (corticotropin releasing hormone, CRH), SCN (vasoactive intestinal peptide (VIP), arginine vasopressin (AVP), gastrin-releasing peptide (GRP)), or pineal gland (e.g., melatonin)[10,12,101–104], this study reveals large-scale diurnal changes in ChP translation that lead to altered CSF contents. Additionally, metabolism and mitochondrial function are disrupted in individuals also at

high risk for psychosis[105,106] and bioenergetics profiles are altered during late onset Alzheimer's disease[107]. The current study defines a proteostatic and metabolic framework in the ChP that changes diurnally, and in the case of TTR, in response to systemic cues like feeding. These data open avenues for future studies to identify whether such metabolic and CSF composition changes are causes or symptoms of neurological disease. Specifically, our results showing the ability of behavioral interventions to alter the CNS fluid environment represent an initial step to identifying daily CSF variation and suggest methods of controlling its composition.

To generate these tools, we overcame several challenges and encountered new limitations. (1) To enrich for actively translated transcripts, we used the TRAP ribosomal pulldown technique. However, TRAP has a preference for capturing changes in metabolic translation, as ribosomes containing RPL10A preferentially translate metabolic, cell cycle, and developmental targets[30]. Transcripts that are preferentially translated by other ribosomal subunits are not part of this analysis. (2) Circadian transitions are continuous, but we performed the initial discovery stages using TRAP ribosome pulldown at only 2 orthogonal timepoints, and therefore we cannot rule out the possibility that some important circadian ChP protein changes were not discovered using this method. (3) This study emphasized changes in protein production, but quick circadian shifts in protein availability will also require analysis of protein degradation and turnover.

## Methods

This research complies with all relevant ethical regulations and was approved by Boston Children's Hospital IACUC and Boston Children's Hospital Institutional Biosafety Committee.

### Mice

The Boston Children's Hospital IACUC approved all experiments involving mice in this study. Adult CD1 male mice were obtained from Charles River Laboratories. Mice with germline Tg(FOXJ1-cre) F26Htzm (referred as *Foxj1-cre*) were imported from Washington University[31] and bred in-house. Transgenic floxed EGFP-L10a mice [B6;129S4-Gt(ROSA)26Sor^tm9(EGFP/Rpl10a)Amc/J], *Bmal1*-null mice [B6.129-Arntl^tm1Bra/J][40] and *Per2-Luciferase* mice [B6.129S6-Per2^tm1Jt/J] were purchased from Jackson Laboratories (Stock Numbers: 024750, 009100 and 006852, respectively). The *Ttr^mNeonGreen* mouse line was generated as detailed below. Animals were housed in a temperature- and humidity-controlled room (70 ± 3 °F, 35–70% humidity) on a 12 h light/12 h dark cycle (7 a.m. on 7 p.m. off) and had free access to food and water, unless otherwise specified. All mice younger than postnatal day 10 were allocated into groups based solely on the gestational age without respect to sex (both males and females were included). For studies involving mice older than 10 days, only male mice were included.

## TRAP

*Foxj1*:Cre x floxed EGFP-L10a transgenic lines were crossed to generate mice heterozygous for each transgene *Foxj1*:Cre[+/−]; floxed EGFP-L10a[+/−]. Mice were kept in standard housing with 12 h light/ 12 h dark cycle (7 a.m. on/ 7 p.m. off), fed ad libitum, and aged 8 weeks. For TRAP, fresh brain tissue was dissected ($N = 3$ at each time − 9 a.m. and 9 p.m., each $N$ included LV ChP pooled from 3 mice). Whole ChP from the lateral ventricle was harvested using #5 forceps and a scalpel in 1× HBSS. To collect the LV ChP, the cerebellum was separated from the mid- and forebrain using a scalpel, followed by a bilateral cut along the midline to separate the brain into two hemispheres. Each hemisphere was stabilized with forceps and a third of the rostral end was cut off. The medial cortex was removed with a scalpel to expose the ventricle and free margin of the ChP. The attached LV ChP was gently separated from the hippocampus/fornix using forceps and immediately extracted for TRAP RNA purifications[28]. RNA quality was assessed using Bioanalyzer Pico Chips (Agilent, 5067-1513) and quantified using Quant-iT RiboGreen RNA assay kit (Thermo Fisher Scientific R11490). Libraries were prepared using Clonetech SMARTer Pico with ribodepletion and Illumina HiSeq to 50NT single-end reads. Sequencing was performed at the MIT BioMicroCenter.

## Sequencing data analysis

The raw fastq data of 50-bp single-end sequencing reads were aligned to the mouse mm10 reference genome using STAR 2.4.0 RNA-Seq aligner[108]. The mapped reads were processed by htseq-count of HTSeq software[109] with mm10 gene annotation to count the number of reads mapped to each gene. The Cuffquant module of the Cufflinks software[110] was used to calculate gene FPKM (Fragments Per Kilobase of transcript per Million mapped reads) values. Gene differential expression test between transcripts associated with RPL10a and transcripts not associated with RPL10a was performed using DESeq2 package[111] and differentially expressed genes were defined as DeSeq2 adjusted $p < 0.05$; $|\log_2 FC| > 0.3$. To cast a wider net for gene differential test between animal groups at different times, differentially expressed genes identification was performed using CuffDiff module of Cufflinks software[110] and differentially expressed genes were defined as CuffDiff; adjusted $p < 0.05$; $|\log_2 FC| > 0.4$. All tests were done with the assumption of negative binomial distribution for RNA-Seq data. All analyses were performed using genes with FPKM > 1, which we considered as the threshold of expression (Figs. 1, 2 source data). Hierarchical clustering of z-scores reveals consistency among all three samples (Fig. 2).

## Sequencing pathway and motif analysis

Functional annotation clustering was performed using DAVID v6.7[112]. Gene ontology (GO) analysis was performed using AdvaitaBio iPathway guide V.v1702. Enrichment vs. perturbation analysis was performed by AdvaitaBio iPathway guide V.v1702 and allows comparison of pathway output perturbation and cumulative gene-set expression changes. In brief, the enrichment analysis is a straightforward gene-set enrichment over representation analysis (ORA) considering the number of differentially expressed genes (DEGs) that are assigned to a given pathway. The enrichment value is expressed as a proportion of enriched members to total genes in a defined pathway and a *p*-value (Fisher) is calculated for this score[113]. Motif analyses were performed using SignalP (v5.0)[114,115] and TMHMM (v2.0)[116]. Predicted secreted gene products were validated by Human Protein Atlas and UniProtKB/Swiss-Prot.

## Seahorse metabolic analysis

ChP explants were dissected in HBSS (Fisher, SH30031FS) and maintained on wet ice until plated. Only the posterior leaflet of the LV ChP was retained for analysis due to empirically determined limitations of the oxygen availability in the XFe96 Agilent Seahorse system. Tissue explants were plated on Seahorse XFe96 spheroid microplates (Agilent, 102905-100) coated with Cell TAK (Corning), in Seahorse XF Base Medium (Agilent, 102353–100) supplemented with 0.18% glucose, 1mM L-glutamine, and 1 mM pyruvate at pH7.4 and incubated for 1 h at 37 °C in a non-CO₂ incubator. Extracellular acidification rates (ECAR) and oxygen consumption rates (OCR) were measured via the Cell Mito Stress Test (Agilent, 103015-100) with a Seahorse XFe96Analyzer (Agilent) following the manufacturer's protocols. Data were processed using Wave software (Agilent). ATP production was calculated as the difference in OCR measurements before and after oligomycin injection, as described by the manufacturer's protocol (Agilent, 103015-100). Calcein AM was used to normalize between wells (Invitrogen L-3224).

## Sample preparation for LC-MS analysis of polar metabolites from ChP

Mice (CD1 males) kept in circadian cabinet housing with 12-h light cycle (7 a.m. on/ 7 p.m. off) aged 8 weeks ($N = 16$ at each time−9 a.m. and 9 p.m.), were decapitated and brain tissue was immediately dissected and frozen on dry ice. ChP were extracted by brief sonication in 200 μl of extraction solvent (80% LC/MS-grade methanol, 20% 25 mM Ammonium Acetate and 2.5 mM Na-Ascorbate prepared in LC/MS water and supplemented with isotopically labeled internal standards (17 amino acids and isotopically labeled reduced glutathione, Cambridge Isotope Laboratories, MSK-A2-1.2 and CNLM-6245-10). After centrifugation for 10 min at maximum speed on a benchtop centrifuge (Eppendorf) the cleared supernatant was dried using a nitrogen dryer and reconstituted in 20 μl water (supplemented with QReSS, Cambridge Isotope Laboratories, MSK-QRESS-KIT) by brief vortexing. Extracted metabolites were spun again and cleared supernatant was transferred to LC-MS micro vials. A small amount of each sample was pooled and serially diluted 3- and 10-fold to be used as quality controls throughout the run of each batch.

## Sample preparation for LC-MS analysis of thyroid hormone metabolites from CSF and plasma

CSF was collected from adult (3 months old) wild-type CD1 mice. Samples were placed on wet ice, then spun $1000 \times g$ for 10 min at 4 °C. The supernatant was collected. Per condition, 5–10 μL of fresh, cleared CSF was extracted in 4:6:3 chloroform:methanol:water mixture supplemented with isotopically labeled T3 and T4 (Cambridge Isotope Laboratories, CLM-7185-C and CLM-8931-PK) as well as isotopically labeled 17 amino acids and isotopically labeled reduced glutathione (Cambridge Isotope Laboratories, MSK-A2-1.2 and CNLM-6245-10). After centrifugation for 10 min at maximum speed on a benchtop centrifuge (Eppendorf) the top, hydrophilic layer was transferred to a new tube, dried using a nitrogen dryer and reconstituted in 20 μl water (supplemented with QReSS, Cambridge Isotope Laboratories, MSK-QRESS-KIT) by brief vortexing. Extracted metabolites were spun again and cleared supernatant was transferred to LC-MS micro vials. A small amount of each sample was pooled and serially diluted 3- and 10-fold to be used as quality controls throughout the run of each batch.

## Chromatographic conditions for LC/MS

**ZIC-pHILIC chromatography for polar metabolites.** 1−2 μl of reconstituted sample was injected into a ZIC-pHILIC 150 × 2.1 mm (5 μm particle size) column (EMD Millipore) operated on a Vanquish™ Flex UHPLC Systems (Thermo Fisher Scientific, San Jose, CA). Chromatographic separation was achieved using the following conditions: buffer A was acetonitrile; buffer B was 20 mM ammonium carbonate, 0.1% ammonium hydroxide. Gradient conditions were as follows: linear gradient from 20 to 80% B; 20–20.5 min: from 80 to 20% B; 20.5–28 min: hold at 20% B. The column oven and autosampler tray were held at 25 °C and 4 °C, respectively.

**C18 chromatography for T3/T4.** 5–7 μl of reconstituted sample was injected onto an Ascentis Express C18 HPLC column (2.7 μm × 15 cm × 2.1 mm; Sigma Aldrich). The column oven and autosampler tray were held at 30 °C and 4 °C, respectively. The following conditions were used to achieve chromatographic separation: buffer A was 0.1% formic acid; buffer B was acetonitrile with 0.1% formic acid. The chromatographic gradient was run at a flow rate of 0.250 ml min⁻¹ as follows: 0–5 min: gradient was held at 5% B; 2–12.1 min: linear gradient of 5 to 95% B; 12.1–17.0 min: 95% B; 17.1–21.0 min: gradient was returned to 5% B.

## MS data acquisition conditions for targeted analysis of polar metabolites and thyroid hormones

MS data acquisition was performed using a QExactive benchtop orbitrap mass spectrometer equipped with an Ion Max source and a HESI II probe (Thermo Fisher Scientific, San Jose, CA) and was performed in positive and negative ionization mode in a range of $m/z = 70–1000$, with the resolution set at 70,000, the AGC target at $1 \times 10^6$, and the maximum injection time (Max IT) at 20 msec. A narrower scan in positive mode at $m/z = 600–800$ was used for more specific detection of thyroxine hormones, the resolution was set at 70,000, the AGC target was $5 \times 10^5$, and the max IT was 100 msec. For polar metabolites HESI conditions were: Sheath gas frow rate: 35; Aug gas flow rate: 8; Sweep gas flow rate: 1; Spray voltage: 3.5 kV (pos), 2.8 kV (neg); Capillary temperature: 320 °C; S-lens RF: 50; Aux gas heater temperature: 350 °C. For T3/T4 HESI conditions were: Sheath gas frow rate: 40; Aug gas flow rate: 10; Sweep gas flow rate: 0; Spray voltage: 3.5 kV (pos), 2.8 kV (neg); Capillary temperature: 380 °C; S-lens RF: 60; Aux gas heater temperature: 420 °C.

Relative quantitation of polar metabolites was performed with TraceFinder 5.1 (Thermo Fisher Scientific, Waltham, MA) using a 5 ppm mass tolerance and referencing an in-house library of chemical standards. Pooled samples and fractional dilutions were prepared as quality controls and only those metabolites were taken for further analysis, for which the correlation between the dilution factor and the peak area was >0.95 (high-confidence metabolites) and for which the coefficient of variation (CV) was below 30%. Normalization for biological material amounts was based on the total integrated peak area values of high-confidence metabolites within an experimental batch after normalizing to the averaged factor from all mean-centered chromatographic peak areas of isotopically labeled amino acids and internal standards. For thyroid hormone peak normalization, isotopically labeled thyroid hormone standards were used. Where indicated, data was control mean-centered, otherwise data were Log transformed and Pareto scaled within the MetaboAnalyst-based statistical analysis platform (v5.0)[117]. All heatmap, PCA, or PLSDA analysis were generated using the MetaboAnalyst online platform. Individual one-way Anova and t-tests were performed in Prism software (GraphPad v7).

## Tissue processing for histology

Samples were fixed in 4% paraformaldehyde (PFA). For cryosectioning, samples were incubated in the following series of solutions: 10% sucrose, 20% sucrose, 30% sucrose, 1:1 mixture of 30% sucrose and OCT (overnight), and OCT (1 h). Samples were frozen in OCT.

## Immunostaining

Cryosections were blocked and permeabilized (0.3% Triton-X-100 in PBS; 5% serum), incubated in primary antibodies overnight and secondary antibodies for 2 h. Sections were counterstained with Hoechst 33342 (Invitrogen H3570, 1:10000) and mounted using Fluoromount-G (SouthernBiotech). The following primary antibodies were used: chicken anti-GFP (Abcam ab13970 (RRID: AB_300798); 1:1000), rabbit anti-pS6 ribosomal protein (Ser 240/244) (Cell Signaling #5364 S (RRID: AB_10694233); 1:1000), rabbit anti-citrate synthase

(cloneD7V8B) (Cell Signaling #14309 (RRID: AB_2665545); 1:1000), mouse anti-fibrillarin (Abcam ab4566 (RRID: AB_304523); 1:250). The following secondary antibodies were used for immunostaining: Goat anti-Rabbit IgG (H + L) Highly Cross-Adsorbed Secondary Antibody, Alexa Fluor 488 (Thermo Fisher A11034; 1:500); Goat anti-Mouse IgG (H + L) Highly Cross-Adsorbed Secondary Antibody, Alexa Fluor Plus 647 (Thermo Fisher A32728; 1:500); Goat anti-Chicken IgY (H + L) Secondary Antibody, Alexa Fluor 488 conjugate (Thermo Fisher A11039; 1:500). Citric acid antigen retrieval was used for fibrillarin staining (15 min steaming in 10 mM sodium citrate, 0.05% Tween 20, pH = 6.0). Secondary antibodies were selected from the Alexa series (Invitrogen/ Thermo Fisher, 1:500). Images were acquired using Zeiss LSM880 confocal microscope with ×20 objective.

## Quantification of nucleolar volume

Nucleolar volume was quantified in accordance with published methods using Imaris in a blinded manner[118–121]. Cryosections for quantification were 20 μm thick and stained with anti-fibrillarin antibody (Abcam, Cambridge, MA). Z-stack images were acquired using the ×40 objective on an LSM 700 laser scanning confocal microscope (Carl Zeiss, Oberkochen, Germany). Three-dimensional reconstruction was performed using the Surface tool in Imaris image analysis software version 7.7.1 (Bitplane, Zurich, Switzerland). Nucleoli with sphericity <0.44 or volume <0.10 μm³ were considered staining artefacts and excluded. Median values of all nucleolar volumes from a single mouse were plotted ($N = 5$ mice).

## OPP incorporation

ChP OPP incorporation was performed as described by Liu et al.[27]. Mice received intraperitoneal OPP injections (50 mg/kg OPP; Life Technologies). One hour later, tissues were obtained and sectioned to a thickness of 7 mm using a cryostat. OPP signals were detected using the Click-iT plus OPP protein synthesis assay kits (Life Technologies) according to the manufacturer's suggested procedures. Images were taken at 20 × (Zeiss Axio Observer D1 inverted microscope) and fluorescence intensity was quantified using FIJI (ImageJ). For each sample, OPP intensity was measured in FIJI in an ROI that included only LV ChP tissue.

## Rhythmicity analysis of RT-qPCR data

RAIN (Rhythmicity Analysis Incorporating Nonparametric methods)[122] was used to analyze rhythmicity of 3-h qPCR data and multi-day TTR:mNeonGreen light–dark and dark-dark photometry data using Bioconductor version 3.10 (BiocManager 1.30.10) in R version 3.6.3 (2020-02-029) with the period set at 24 h.

## Per2- luciferase explant luminometry

ChP explants from *Per2:luc* mice[43] were dissected in HBSS (Fisher, SH30031FS) and maintained on wet ice until plated. Explants were transferred to 24 well plates with 500 μL of filter-sterilized Lumicycle media (phenol-free DMEM (Sigma D-2902), 10 mM HEPES (pH 8.0), 4 mM sodium bicarbonate, 25 mM D-glucose, 1 × B27 (Gibco), 4mM L-glutamine, Penicillin/Streptomycin) freshly supplemented with 100 mM beetle D-luciferin (Promega, E1601) and/or 100 nM dexamethasone. PCR plate sealer was used to seal the wells for the duration of the experiment (ThermoFisher) and placed in a Lumicycle-96 (Actimetrics) in a water-jacketed incubator at 35 ˚C. Luminometry was performed by iterative measurement in 60 s bins, 10 times per hour. Luminometry data was analyzed with Lumicycle analysis software (Actimetrics). Only traces with a goodness-of-fit of at least 80% were included. Period was measured using the $\chi^2$ function.

## Immunoblotting

Tissues were homogenized in RIPA buffer supplemented with protease and phosphatase inhibitors. Protein concentration was

determined by BCA assay (Thermo Scientific 23227). Samples were denatured in 2% SDS with 2-mercaptoethanol by heating at 37 °C (for OxPhos) or 95 °C (for TTR and SERPINE2) for 3 min. Samples were centrifuged at maximum speed and then equal amounts of proteins were loaded and separated by electrophoresis in a 4–15% gradient polyacrylamide gel (BioRad #1653320) or NuPAGE 4–12% Bis-Tris gel (Invitrogen #NP0322), transferred to a nitrocellulose membrane (250 mA, 1.5 h, on ice), blocked in filtered 5% BSA or milk in TBST, incubated with primary antibodies overnight at 4 °C followed by HRP conjugated secondary antibodies (1:5000) for 1 h, and visualized with ECL substrate. For phosphorylated protein analysis, the phospho-proteins were probed first, and then blots were stripped (Thermo Scientific 21059) and reprobed for total proteins. The following primary antibodies were used: rabbit anti-S6 ribosomal protein (Cell Signaling #2217 S (RRID: AB_331355); 1:1000), rabbit anti-pS6 ribosomal protein (Ser 240/244) (Cell Signaling #5364 S (RRID: AB_10694233); 1:1000), rabbit anti-citrate synthase (cloneD7V8B) (Cell Signaling #14309 (RRID: AB_2665545); 1:1000), rabbit-anti-mitofusin2 (clone D2D10) (Cell Signaling #9482 (RRID: AB_2716838); 1:1000), rabbit anti-Vinculin (Cell Signaling #13901 S (RRID: AB_2728768); 1:10000), Total OXPHOS Rodent WB Antibody Cocktail (Abcam ab110413 (RRID: AB_2629281); 1:250), rabbit Anti-human prealbumin (TTR) (Dako/Agilent A000202-2 (RRID: AB_578466); 1:200), mouse anti-transferrin [3C11] (Abcam ab70826 (RRID: AB_1281166); 1:750), mouse anti-mNeonGreen (ChromoTek Cat# 32f6-100 (RRID:AB_2827566); 1:1000). The following secondary antibodies were used for immunoblotting: Goat anti-Mouse IgG (H + L) HRP (Life Technologies 31430; 1:1000); Goat anti-Rabbit IgG (H + L) HRP (Life Technologies 31460; 1:1000). Source data are provided as a Source Data file (Source Data).

## Quantitative RT-PCR

For mRNA expression analyses, the ChP were collected and one quarter of 4 V ChP or one-half of one LV ChP was frozen for RT-qPCR, while the rest of the tissue from an individual was frozen for protein analysis. RNA was isolated using the RecoverAll RNA/DNA isolation kit (Life Technologies A26135) for ChP and using TRIzol Reagent (Invitrogen 15596026) for liver samples following manufacturer's specifications for RNA extraction. Extracted RNA was quantified spectrophotometrically and 50 ng was reverse-transcribed into cDNA using the Promega™ ImProm-II™ Reverse Transcription System (Fisher PR-A3802) following manufacturer's specifications with random hexamer primers (Promega PR-C1181). RT-qPCRs were performed in duplicate using Taqman Gene Expression Assays and Taqman Gene Expression Master Mix (Applied Biosystems) with *Gapdh* or *Rp1p0* as an internal control. Cycling was executed using the StepOnePlus Real-Time PCR System (Invitrogen) and analysis of relative gene expression was performed using the $2^{-\Delta\Delta CT}$ method[123] and presented as RQ of data normalized to 9 am and the internal control. Technical replicates were averaged for their cycling thresholds and further calculations were performed with those means. The following mouse Taqman probes (Thermo Fisher) were used: *Ttr* (Mm00443267_m1), *Serpine2* (Mm00436753_m1), *Bmal1*(Mm00500223_m1), *Per2*(Mm00478099_m1), *Itgb8*(Mm00623991_m1), *Cdh3*(Mm01249209_m1), *Abcf3* (Mm00658695_m1), *Slc7a8*(Mm01318974_m1), *Chmp1b* (Mm04179599_s1), *Gapdh* VIC (Mm99999915_g1), *Rp1p0* VIC (Mm00725448_s1).

## Restricted feeding

Adult mice were singly housed in cages placed in a controlled lighting environment away from the main mouse colony. The lights were programmed to come on at 7 a.m. and turn off at 7 p.m. The mice were acclimatized to the boxes for 5 days with *ad libitum* feeding. Then for 7 days, at each lighting change, mice were either moved between a home cage with water and no food, or a home cage with food and water. For control feeding (night feeding), mice were moved to a food-containing cage at 7 p.m. and mice were moved to food-free cages at 7 a.m. For inverted feeding (day feeding), mice were moved to a food-containing cage at 7 a.m. and mice were moved to food-free cages at 7 p.m. All cage changes took no more than 10 min. Mice always had access to water.

## Generation of *Ttr^mNeonGreen* mouse line

*Ttr^mNeonGreen* mice were generated on C57BL6 background using CRISPR technology with homology-directed repair (HDR). The template HDR construct was synthesized to insert the mNeonGreen sequence at the C-terminus of *Ttr*. The following gRNAs were used:

5′-GGGGTTGCTGACGACAGCCG-3′; 5′-CCCATACTCCTACAGCACCA-3′;

5′-AGAATTGAGAGACTCAGCCC-3′

Generation of the mouse line was carried out by Boston Children's Hospital Mouse Gene Manipulation Core. Mice were screened with 2 sets of PCR primers for correct insertion of mNeonGreen protein. PCR products were sequenced to ensure no mutations during the CRISPR editing process.

PCR primer set 1:

GAGCAAGTGACAGAGTTGCCCT; agtctcTTACTTGTACAGCTCGTCCATGCCCATC

PCR primer set 2:

TCTGGTGGAGGTGGATCCGGAG; GGGAAACGACGGATCGGGGAG

The founder was bred to C57BL6 for 5 generations before experiments were performed. PCR primer for genotyping progeny are:

Gentyping primer set: CAGCCATTGGGTGCCACAAC; CTGTCGTCAGCAACCCCCAGAAT.

## Characterization of *Ttr^mNeonGreen* mice

Immunoblotting of the ChP and CSF: The ChP was dissected, and protein extracted (homogenized in RIPA buffer). 50 µg total protein was loaded to a single well for immunoblotting. Up to 15 µl of CSF was loaded to a single well for immunoblotting. All CSF samples were normalized based on total protein level. The blots were incubated in mNeon antibody (Chromotek 32f6, 1:200, Mouse monoclonal) at room temperature for 15 min before moving to 4 °C overnight. TTR and GAPDH were probed on the same blots after stripping.

Immunohistochemistry: Tissue sections were stained with Hoechst and directly mounted. Endogenous mNeon signal was imaged by confocal microscopy.

CSF fluorescence detection: CSF was collected from adult (3 months old) *Ttr^mNeonGreen* mice and their wild-type littermate controls. 10 µl CSF was diluted into 30 µl with PBS and loaded to a 96-well plate with clear bottom. Blank PBS was included as negative control. Fluorescent signal was measured with 488 nm excitation on a Tecan plate reader.

## CSF fiber photometry recordings from freely-moving mice

Wild-type and *Ttr^mNeonGreen* mice used for in vivo fiber photometry experiments were surgically outfitted with a titanium headpost (0.7 g, H.E. Parmer) and bilateral stainless steel 22XX gauge LV guide cannulae (Microgroup, length: 7 mm). Briefly, mice were anesthetized with 1–4% isoflurane in 100% O2 and breathing rate was maintained at 1 breath per second. Animals were administered dexamethasone (1 mg/kg; intramuscular), meloxicam (5 mg/kg; subcutaneous) and 0.9% saline (1 mL; subcutaneous) pre-operatively. The scalp was removed, and bilateral craniotomies were drilled at stereotaxic coordinates ±1 mm lateral and −0.45 mm posterior to bregma using standard asepsis technique and a standing microscope. A guide cannula was lowered to a depth of −2 mm below bregma in each craniotomy and secured using C&B metabond (Parkell). A headpost was then secured to the skin using Vetbond (3 M) and sealed with C&B metabond. Guide cannulae were plugged with homemade pieces of sterile PE 10 tubing (BD

Intramedic) secured with Kwik-Cast silicone sealant (World Precision Instruments).

At least two weeks after surgery, cannulae were acutely opened in awake mice and aCSF (Tocris, 3525) was infused into each cannula to clear out any remaining brain parenchyma, scar tissue and clots. 30 µL of aCSF was infused into each cannula over 5 min using a microinjection catheter and syringe pump (Kent Scientific Corporation). To prevent excessive damage to the ventricular system, infusions of the left and right cannula were performed two days apart. Mice without optically clear CSF in their guide cannulae following bilateral aCSF infusions were not used for photometry experiments. Two days after the final aCSF infusion, an optic fiber with metal ferrule (400 µm diameter core; multimode; numerical aperture (NA) 0.37; 6.5 mm length; Doric Lenses) was placed in each guide cannula and secured with C&B metabond.

Cannulated wild-type and $Ttr^{mNeonGreen}$ mice were singly housed in standard circadian light–dark boxes and exposed to alternating 12-h light and dark periods or constant darkness, depending on the experimental paradigm. Fiber photometry of the CSF was performed using a FP3001 photometry box (Neurophotometrics) with 415 nm, 470 nm, and 560 nm LED light sources and a CMOS camera for signal collection (green emission channel: 494–531 nm, red emission channel: 586–627 nm). Low-autofluorescence fiber optic cables (2 m long; 400 µm diameter core; 0.37 NA; Doric Lenses) with three branches were coupled to implanted optic fibers with zirconia sleeves (Precision Fiber Products). The coupling point was secured with UV-curable optical adhesive (Flow-It ALC, Pentron) and shielded with black heat shrink tubing. Tethered mice were then allowed to freely move about their cages. Freely-moving CSF photometry recording sessions lasted between 24 and 96 h, during which the CSF was exposed to the following sequence of excitation wavelengths: 415 nm (0.2–0.3 mW), 470 nm (0.2–0.3 mW), and darkness (background measurement) at 5 Hz. Signal collection was controlled using the Bonsai Visual Reactive Program (Open Ephys).

### Fiber photometry data analysis
Photometry data from the green channel at 470 nm excitation were preprocessed with a median filter in bins of 10 min. For analyses in the first 24 h of each recording, a stretched exponential fit of form $\alpha + \beta e^{-\gamma t^{\delta}}$ was subtracted from each trace to correct for photobleaching. For analyses beyond the first 24 h, no photobleaching correction was performed. $\frac{\Delta F}{F_0}$ was calculated relative to the first hour of each plot, and $F_0$ was taken as the raw signal prior to photobleaching correction (if it had been performed).

### Mouse computed tomography (CT) imaging and analysis
CT scans of the head were obtained to confirm adequate placement of guide cannulae in bilateral LVs. Mice in the second post-operative week were anesthetized with 1–4% isoflurane (in 100% $O_2$) in an induction chamber and moved to a heated imaging bed in a SPECT/PET/CT scanner (Bruker/Albira). The animal was aligned within the scanner, and CT images of the head and neck were acquired. CT images were then imported into VivoQuant (Invicro, 2021) for analysis. Because it is difficult to distinguish specific brain regions on CT scans, a 3D Brain Atlas Tool in VivoQuant was used to manually superimpose a three-dimensional, virtual projection of the mouse brain onto our images. Overlap between the projected lateral ventricles and bilateral guide cannulae was then confirmed.

### HRP injection and TEM
Adult mice were singly housed in cages placed in a circadian cabinet controlled lighting environment away from the main mouse colony. The lights were programmed to come on at 7 a.m. and turn off at 7 p.m. The mice were acclimatized to the boxes for 5 days with *ad libitum* feeding. Mice were weighed and injected intravenously

through the tail vein with 0.5 mg horseradish peroxidase (HRP-type II, Fisher Scientific PI31491) per g bodyweight diluted such that each mouse received 100uL per 25 g bodyweight. Allow to circulate for 7–10 min, brains were harvested, frontal pole was removed, and fixed 1 h in Karnovsky's fixative, [5% glutaraldehyde, 4% PFA, 0.4% CaCl2, in 0.1 M cacodylate buffer - EMS (Electron Microscopy Sciences], then fixed in 4% PFA in 0.1 M cacodylate at 4 °C overnight while rocking. Brains were coronally sectioned in 200 µm vibratome sections. Those sections containing ventricle and ChP were selected and processed for DAB (Millipore Sigma D5905). Sections were first washed with 20 mM glycine, washed 3 times with cold 0.1 M cacodylate, and developed in filter-sterilized 5 mg DAB in 9 ml of 0.1 M cacodylate plus ~9 mM $H_2O_2$ on ice for 30–45 min until dark brown. The portion containing LV ChP was processed, sectioned, and imaged at the Conventional Electron Microscopy Facility at Harvard Medical School. Tissue was postfixed with 1% osmiumtetroxide (OsO4)/1.5% potassium ferrocyanide (KFeCN6) for one hour, washed in water three times and incubated in 1% aqueous uranyl acetate for one hour. This was followed by two washes in water and subsequent dehydration in grades of alcohol (10 min each; 50%, 70%, 90%, 2 × 10 min 100%). Samples were then incubated in propyleneoxide for one hour and infiltrated overnight in a 1:1 mixture of propyleneoxide and TAAB Epon (Marivac Canada Inc. St. Laurent, Canada). The following day, the samples were embedded in TAAB Epon and polymerized at 60 degrees C for 48 h. Ultrathin sections (about 80 nm) were cut on a Reichert Ultracut-S microtome, and picked up onto copper grids stained with lead citrate. Sections were examined in a JEOL 1200EX Transmission electron microscope or a TecnaiG² Spirit BioTWIN. Images were recorded with an AMT 2k CCD camera.

### Epithelial junction analyses
Ten images of epithelial junctions were acquired for each animal. $N = 3$ animals at each time. Images were acquired at ×2000 for structure. Analysis was performed on tiled ×10,000 images reassembled by hand in Illustrator (Adobe). A peri-junctional zone was defined as the 2 µm wide zone along the lateral edge of 2 ChP epithelial cells. Electron-dense HRP-filled vesicles were counted and recorded in FIJI ImageJ (NIH). Apical-basal localization was calculated using a custom MatLab (Mathworks; R2019b) code as before[124]. Tight junction (TJ) was defined by the apico-lateral area where plasma membranes make complete contact accompanied by continuous, anastomosing network of intra-membranous particle strands (TJ strands or fibrils). For each TJ, 10 measurements were made between the 2 plasma membranes within the junction that were not in direct contact along the full apico-basal extent of the junction. These 10 measurements were averaged to obtain the width of each junction. Ten images were analyzed for each animal. The average of these 10 averages was plotted for each $N$. $N = 3$ animals at each time.

### Ex vivo ChP 3-day imaging
After dissection in 1× HBSS, LV ChP from a TTR m:Neon mouse was transferred onto 35 mm glass bottom imaging dishes. (MatTek, Cat. P35G-1.5-14-C) that had been prepared as follows: briefly, the edges of the glass bottom dish were lightly ringed with Silicone (Kwik-sil, World Precision Instruments, Item. 600022), and a polycarbonate membrane (Whatman, Nucleopore, 13 mm wide, 8.0 mm pore size, Cat. 110414) with hole cut out of the center was placed on the glass. These dishes were kept at room temperature and allowed to cure. Dishes were filled with 4 mL of filter-sterilized Lumicycle media exclusive of the luciferin (phenol-free DMEM (Sigma D-2902), 10 mM HEPES (pH 8.0), 4 mM sodium bicarbonate, 25 mM D-glucose, 1× B27 (Gibco), 4mM L-glutamine, Penicillin/Streptomycin). The ChP was flattened onto the glass bottom dish and secured to the membrane using 3 M Vetbond. All samples were placed in a standard incubator at 37 °C until imaging commenced. Explants were imaged in an oxygenated/CO₂-treated,

heated at 37 °C, humidified chamber on a Leica DMi8 microscope. Once region was identified, tissue was synchronized with 100 nM dexamethasone. Three stacks of 40 z-steps were acquired for each timepoint and averaged. One set of 3 stacks was acquired every 30 min for 72 h using the HC PL fluotarL 20x/0.40 dry objective and Leica-DFC9000GT-VSC07354 camera. To reduce photobleaching, 460 nm LED power was kept under 18%.

Prior to analysis, each stack of 3 scans was averaged together, and then binned in XY by a factor of 2. A local entropy filter with width of 10 pixels was applied to each plane of the resulting stack using the MATLAB function `entropyfilt`, creating an entropy volume with greater pixel values in high-detail regions. These high entropy areas correspond to regions in which the cell body lattice is in focus. This entropy volume was max projected along the Z-axis, and the linear Z index is recorded for an in-focus map. This map was smoothed using a Gaussian filter with width of 20 pixels, and applied to the original volume, resulting in a focused projected image. This image was calculated at each 30 min time period for 72 h to make a focused video.

The focused video was then high-pass filtered and roughly registered, first with a rigid registration followed by an affine registration. Both registrations were performed using the `StackReg` algorithm in ImageJ. To account for slow-scale warping of the tissue over the course of 72 h, a local non-linear registration was then applied by calculating the corresponding displacement field between each movie frame using the `imregdemons` algorithm in MATLAB. For each non-linear registration, the mean of all frames was used as a reference target.

This focused and registered image was cropped to the center 50% in both X and Y, and the pixel values were rescaled between the 1st and 99th percentile to normalize the fluorescence change among trials. The fluorescent signal was finally calculated by averaging all normalized pixel values over time.

### Quantification and statistical analysis

Biological replicates (*N*) were defined as samples from distinct individual animals, analyzed either in the same experiment or within multiple experiments, with the exception when individual animal could not provide sufficient sample (i.e., CSF), in which case multiple animals were pooled into one biological replicate and the details are stated in the corresponding figure legends. Statistical analyses were performed using Prism (GraphPad) v7 or R (version 3.6.3). Outliers were excluded using ROUT method ($Q = 1\%$). Appropriate statistical tests were selected based on the distribution of data, homogeneity of variances, and sample size. The majority of the analyses were done using One-way ANOVA or Student's two-tailed unpaired *t* test (when *F* test failed to discover unequal variances) or Welch's two-tailed unpaired *t* test (when *F* test discovered unequal variances), except for Fig. 3d and Supplementary Fig. 2c where the analysis was done using Kolmogorov–Smirnov tests. *F* tests or Bartlett's tests were used to assess homogeneity of variances between datasets. Parametric tests (*T* test, ANOVA) were used only if data were normally distributed, and variances were approximately equal. Otherwise, nonparametric alternatives were chosen. Data are presented as means ± standard deviation (SD). If multiple measurements were taken from a single individual, data are presented as means ± standard errors of the mean (SEMs). Please refer to figure legends for sample size. *p* values < 0.05 were considered significant (\**p* < 0.05, \*\**p* < 0.01, \*\*\**p* < 0.001, \*\*\*\**p* < 0.0001). Exact *p* values can be found in the figure legends. *P* values are also marked in the figures where space allows.

### Data availability

The sequencing data from the TRAP study (for Figs. 1–3) generated in this study have been deposited in the GEO database under accession number GSE161877. The metabolomics data have been deposited in the Metabolomics Workbench database[125] under study ID ST002707. Raw and processed image and quantification data are available at the corresponding author's website, upon request, or are provided in Supplementary Information. Source data are provided with this paper.

### Code availability

Code used for photometry processing, time-lapse explant imaging analysis, and vesicle apical-basal localization (Figs. 4 and 6) is publicly available at https://github.com/LehtinenLab/diurnal-csf [https://doi.org/10.5281/zenodo.7933530].

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

## Acknowledgements

We thank members of the Lehtinen, Heiman, Kanarek, Lipton, and Andermann labs for helpful discussions and experimental advice; Kevin Chau for advice regarding OPP and nucleoli analysis; Shachar Dagan, Hannah Zucker, and Chenghua Gu for detailed technical and analytical advice regarding HRP injection; Nancy Chamberlin for critical reading of the manuscript; the MIT BioMicro Center for TRAP sequencing support; Harvard Medical School Electron Microscopy Facility for performing TEM; Boston Children's Hospital Mouse Gene Manipulation Core for generating the *Ttr*^*mNeonGreen* mice; Harvard Digestive Diseases Center Cell Function and Imaging Core for supporting Seahorse experiments; Nathaniel Hodgson and Michela Fagiolini with the Boston Children's Hospital Animal Behavior and Physiology Core for supplying circadian cabinets and photometry rigs; Dan Taub from Clifford Woolf's lab for troubleshooting photometry setup; Alan Packard and Andrew LaBella with the Boston Children's Hospital Small Animal Imaging Lab (SAIL) Core for supporting CT imaging experiments. We are grateful for the following support: NIH T32 HL110852 and OFD/BTREC/CTREC Faculty Development Fellowship Award (R.M.F.); NIH F30 DK131642, T32 HL007901, NIH NIGMS T32 GM007753, NIH NIGMS T32 GM144273 (P.N.K.); NIH R35-HL145242 and R01-AI130591 (M.J.H.); NIH R01 HL151368 and DoD W81XWH-18-1-0194 (J.O.L.); JPB Foundation (M.H.); NIH grant U2C-DK119886 and OT2-OD030544 (Metabolomics Workbench); BCH Pilot Grant, Human Frontier Science Program (HFSP) research program grant #RGP0063/2018, NIH R01 NS088566, NIH RF1 DA048790, and the New York Stem Cell Foundation (M.K.L.); and BCH IDDRC 1U54HD090255. The content is solely the responsibility of the authors and does not necessarily represent the official views of the National Institute of General Medical Sciences or the National Institutes of Health.

## Author contributions

R.M.F., P.N.K., P.A.S., H.X., B.P., M.L.A., N.K., J.O.L., and M.K.L. designed and performed experiments; R.M.F., M.L.S., J.P.H., F.B.S., P.N.K., P.A.S., B.P., M.L.A., N.K., J.O.L., and M.K.L. analyzed the data; Y.Z., and M.J.H. provided material; A.V., F.G., P.A.S., B.G., N.D., J.P.H, S.G., A.P, F.C., H.H.-M., and M.H. provided technological support; R.M.F. and M.K.L. wrote the manuscript. All co-authors edited the manuscript.

## Competing interests
The authors declare no competing interests.
