## [Peer Review File · Nature Communications]

Defining diurnal fluctuations in mouse choroid plexus and CSF at high molecular, spatial, and temporal resolutionREVIEWER COMMENTS

Reviewer #1 (Remarks to the Author):

The manuscript # NCOMMS-22-51174 by Fame et al., uses a variety of techniques to evaluate broad diurnal changes in transcript abundance, transcript ribosomal association and protein levels in the murine choroid plexus. The authors identify several key proteins, hormones and cell biological processes that are diurnally regulated in the choroid plexus during the circadian cycle and in response to feeding behavior. Finally, the authors establish an intravital CSF fiber photometry to track ChP output and demonstrate very nicely at the functional level that the choroid plexus undergoes diurnal control in secretion of proteins. Overall, the study is performed at a very high technical level and the transcriptome or proteome studies are very nicely validated for a large number of pathways identified to be regulated by the circadian cycle in the choroid plexus leading to composition of the CSF. There are a few minor concerns that would strengthen the manuscript further as follows:

Minor Concerns:

1. In Figure 6 and S6 the authors examine the diurnal variation of genes associated with the blood-CSF barrier namely adherens and tight junctions and transcytosis. Although the transcripts for several of these are changed between day and night in the choroid plexus (S6), the authors do not find any evidence that the number of HRP vesicles is different. However, they claim that there is an increased width of tight junctions. The TEM data are somewhat unconvincing. The HRP vesicles are difficult to see in a colored TEM image and the authors need to provide images both at day and night. The TEM images of tight junctions are unconvincing since the tight junctions look fine to me. The authors point to the start of the junction as abnormal, but in fact that region is the adherens, not the tight junction. The authors need to measure the length or the curvature of the tight junctions. Since they have injected HRP in these mice, it may be best to see if HRP is found between the junctions. As an alternative, the authors should perform some functional studies with a very small tracer (Sodium fluorescein) in combination with their intravital CSF fiber photometry imaging to determine if indeed there are differences in the BCSFB permeability regulated by tight junctions at day and night. These experiments will be more convincing to demonstrate diurnal fluctuation in BCSFB function.
2. In Figure 5 and S5, the authors examine the diurnal variation of genes associated with the oxidative phosphorylation in mitochondria. Although, the authors demonstrate that several genes of the oxidative phosphorylation are reduced in the evening, there is no difference in seahorse activity in the mitochondria in vitro. Can the authors explain this discrepancy? Is this due to the culture conditions in vitro? Perhaps the authors can explain this better in the discussion.

Reviewer #2 (Remarks to the Author):

This is a well-conducted study, which addresses an important issue in the field. I have no comments.

Reviewer #3 (Remarks to the Author):

The current manuscript entitled "Defining diurnal fluctuations in the 1 choroid plexus and CSF at high molecular, spatial, and temporal resolution" by Fame et al., presents novel data by which the mouse choroid plexus is functionally regulated by circadian rhythms. Authors use a number of advanced techniques to measure gene expression, protein levels, and metabolites associated with diurnal changes in mice. All data is technically sound and provides new information in choroid plexus function.

This reviewer has only minor comments to improve the clarity of the manuscript.

Authors use several techniques to verify changes in protein translation occurring during the day or night (9pm vs 9am) in the CP that are consistent with the liver. Question- why the liver? And not the SCN?

Fig 1h: quantification of nucleolar volume, n=3 each time but many more points on graph. Data should

be presented per animal unless demonstrating within animal variability.

This statement “This overall analysis of ChP translation protein synthesis revealed meaningful differences between the dark and light phases in a Bmal1-dependent manner, with higher levels of protein synthesis during the dark phase” is unfounded. Authors provide no experimental evidence that this is dependent on Bmal1- ie., Bmal1 KO experiment.

Line 197- please indicate that supp fig 3j is the liver to be clear.

Figure 3m and supp fig 3l- it is unclear why different timepoints were chosen here. Previously 9AM and 9PM were being compared and now 5AM-11AM and 5PM-11PM. Authors should at least indicate the reasoning for this.

Line 321-322: Slc9a8 and Nrf2 mentioned to be in fig 6a-b, S6a are not- were these not measured? Or not significant in this dataset compared to previously reported studies references? Adding this information would be useful.

Fig S6a: Itgb8 is presented twice in colour map.

Reviewer #4 (Remarks to the Author):

This study identified differentially expressed genes during diurnal changes by ribosome profiling of ChP epithelial cells. With identified genes, the authors identified several biological processes that related to diurnal variations in mouse ChP and CSF. The findings can be very important to this field. However, there are many concerns need to be addressed:

Major concerns:

1. On line 107-109: the author over claimed that “Overall, translation was increased during the dark phase and diurnal variation in translation was dependent on systemic Bmal1 expression.” based on the data they presented on Fig1. They need to do more to make the conclusion solid, for example, Per2 expression, nucleolar volume in LV ChP of Bmal1 ^{-/-} mice.
2. The author claimed that choroid plexus translation is diurnally regulated. Ribosome biogenesis is one readout for increased translation capacity and any gene related to ribosome may affect ribosome biogenesis. Unfortunately, Rpl10a is one of the ribosome genes and TRAP mouse line is based on Cre-dependent overexpression of Rpl10a from Rosa26 locus. The author should have avoided TRAP mouse line to profile gene expression. Besides, the genotype of L10aGFP (homozygous or heterozygous) in the Foxj1-Cre::L10aGFP line is also expected to affect ribosome biogenesis since it can be one allele overexpression of Rpl10a or both-alleles overexpression of Rpl10a. Another mouse line, RPL22ha(Ribotag), can be a better choice, because the Rpl22-HA expression is much more “endogenous”.
3. on line 129-131: “Analyses of all ChP transcripts associated with ribosomes revealed 779 differentially translated transcripts at 9 a.m. vs. 9 p.m.: 431 enriched at 9 a.m., and 348 enriched at 9 p.m.” However, from figure 2a, I realize that the sample 3 from group “9AM” is grouped with sample 1 from group “9PM”, also, even in the same group, there are big variations. The data quality is worrisome. To make the data solid, the author may consider adding more “N”.
4. To pick top differentially expressed genes, it would be better to use a volcano plot.
5. Figure 6b, all the gene expression is at transcription level, not protein level, also, many of them show big variations. Figure 6g-h, three samples per group is too few. The data is not convincing enough, need more N.

Minors:

1. On line 57: “The choroid plexus (ChP) is a key source of CSF. The specialized ChP epithelial cells that”: the author should had given the full name of ChP in the ABSTRACT.
2. Figure 1d, the author need to show the first timepoint “mid” better, it looks like the “mid” data was plot on Y-axis.

REVIEWER COMMENTS

Reviewer #1 (Remarks to the Author):

The manuscript # NCOMMS-22-51174 by Fame et al., uses a variety of techniques to evaluate broad diurnal changes in transcript abundance, transcript ribosomal association and protein levels in the murine choroid plexus. The authors identify several key proteins, hormones and cell biological processes that are diurnally regulated in the choroid plexus during the circadian cycle and in response to feeding behavior. Finally, the authors establish an intravital CSF fiber photometry to track ChP output and demonstrate very nicely at the functional level that the choroid plexus undergoes diurnal control in secretion of proteins. Overall, the study is performed at a very high technical level and the transcriptome or proteome studies are very nicely validated for a large number of pathways identified to be regulated by the circadian cycle in the choroid plexus leading to composition of the CSF. There are a few minor concerns that would strengthen the manuscript further as follows:

We thank the reviewer for their careful reading of our manuscript, support for its publication in Nature Communications, and helpful recommendations to improve the study before publication. We address each of the reviewer's comments below.

Minor Concerns:

1. In Figure 6 and S6 the authors examine the diurnal variation of genes associated with the blood-CSF barrier namely adherens and tight junctions and transcytosis. Although the transcripts for several of these are changed between day and night in the choroid plexus (S6), the authors do not find any evidence that the number of HRP vesicles is different. However, they claim that there is an increased width of tight junctions. The TEM data are somewhat unconvincing. The HRP vesicles are difficult to see in a colored TEM image and the authors need to provide images both at day and night. The TEM images of tight junctions are unconvincing since the tight junctions look fine to me. The authors point to the start of the junction as abnormal, but in fact that region is the adherens, not the tight junction. The authors need to measure the length or the curvature of the tight junctions. Since they have injected HRP in these mice, it may be best to see if HRP is found between the junctions. As an alternative, the authors should perform some functional studies with a very small tracer (Sodium fluorescein) in combination with their intravital CSF fiber photometry imaging to determine if indeed there are differences in the BCSFB permeability regulated by tight junctions at day and night. These experiments will be more convincing to demonstrate diurnal fluctuation in BCSFB function.

*Thank you for pointing out these sources of confusion. After consulting with multiple TEM and ChP experts, we maintain that the images shown and measured are indeed tight junctions (TJs), not adherens junctions. However, we realize that the images did not give sufficient context for the readers to evaluate this question. We have now better contextualized the example TEM images to highlight the junction location with respect to the full ChP epithelial cell in **Fig. 6g** with*

additional examples in **Supp. Fig 6e**. Secondly, we did not mean to not claim that the barrier is dysfunctional, only more open, as has been quantified in other publications (Tao-Cheng, Nagy, and Brightman J. Neurosci. 1986). We have now updated the text to more precisely outline our conclusions. Third, on curvature and length: curvature of TJs has been shown to be variable under the same conditions indicating their natural flexibility (Zhao...Kachar 2018 Communications Biology) and while length represents TJ continuity, it is more appropriately quantified in fractured EM, not sectioned TEM. Width, which we quantified, represents the depth of the diffusion barrier (Tao-Cheng, Nagy, and Brightman J. Neurosci. 1986). Finally, the functional assay using CSF photometry is an intriguing one, but is limited by the fact that any change in BBB or B-CSF-B permeability could result in increased tracer access to CSF. Since BBB permeability change is associated with circadian variations (Cuddapah, et al 2019; Pulido, et al 2021), we hesitate to claim ChP specificity from CSF photometry without a ChP-specific intervention.

2. In Figure 5 and S5, the authors examine the diurnal variation of genes associated with the oxidative phosphorylation in mitochondria. Although, the authors demonstrate that several genes of the oxidative phosphorylation are reduced in the evening, there is no difference in seahorse activity in the mitochondria in vitro. Can the authors explain this discrepancy? Is this due to the culture conditions in vitro? Perhaps the authors can explain this better in the discussion.

Thank you for encouraging us to explain our interpretation of these results more fully. We presume that the Seahorse media, which contains high glucose, overrode the intrinsic metabolic activity of the ChP tissue, especially since we later found that at least some ChP circadian properties are feeding-dependent. This is why subsequent ex vivo analyses were performed on purposefully synchronized tissue and monitored for multiple days. In the original version of the manuscript, we noted the following in the results section:

“While we identified large changes in metabolic components from serial samples of acutely collected ChP, we did not observe functional metabolic differences in oxygen consumption or ATP production between 9 a.m. and 9 p.m. in ChP explants in constant 0.18% glucose (Supplementary Figure 5d-e). For reference, normal mouse blood sugar is 80-100 mg/dL (0.08-0.1%) between fasting and feeding, and normal CSF glucose is usually ~60% of the plasma level. We then calculated ChP basal metabolism and ATP production (Supplementary Figure 5e). We presume the that explanted ChP tissue rapidly adapted to the glucose-rich media of the assay, precluding accurate testing of diurnal ChP respiration ex vivo.”

We have now addressed this topic more fully in the Discussion as well.

Reviewer #2 (Remarks to the Author):

This is a well-conducted study, which addresses an important issue in the field. I have no comments.

We thank the reviewer for their careful reading of our manuscript and enthusiastic support for its publication.

Reviewer #3 (Remarks to the Author):

The current manuscript entitled “Defining diurnal fluctuations in the 1 choroid plexus and CSF at high molecular, spatial, and temporal resolution” by Fame et al., presents novel data by which the mouse choroid plexus is functionally regulated by circadian rhythms. Authors use a number of advanced techniques to measure gene expression, protein levels, and metabolites associated with diurnal changes in mice. All data is technically sound and provides new information in choroid plexus function. This reviewer has only minor comments to improve the clarity of the manuscript.

We thank the reviewer for their careful reading of our manuscript, confidence in our study, and helpful recommendations to improve the study before publication. Each comment has been addressed individually below.

Authors use several techniques to verify changes in protein translation occurring during the day or night (9pm vs 9am) in the CP that are consistent with the liver. Question- why the liver? And not the SCN?

Thank you for encouraging us to more fully explain our reasoning. We have included discussion of this point in the Results section. Briefly, circadian changes in translation and metabolism have been robustly defined in the liver. Liver is both strongly impacted by the SCN and by the nutritional clock (whereas SCN is driven predominantly by light). Further, the effects of metabolism on the SCN are still extremely complex and poorly understood. There may indeed be a separate food-entrainable oscillator that is SCN-independent and in that case, SCN would not be the appropriate control for these studies. Therefore, by using the liver as a comparator and positive control for the full circadian rhythmicity of the mice, we were more likely to capture an integrated (i.e. metabolic plus light-driven) rhythm that we then subsequently investigate through inverted feeding (metabolic clock) and Bmal1 null (intrinsic clock) analyses.

Fig 1h: quantification of nucleolar volume, n=3 each time but many more points on graph. Data should be presented per animal unless demonstrating within animal variability.

*The method for presenting nucleolar volume quantification that we apply here has been used in previous publications (e.g. Sanchez, et al Cell Stem Cell 2016; Chau, et al eLife 2018), but we agree that plotting biological replicates is more appropriate. We now add more N (N=5 mice) and also present data points of the median nucleoli volume from each animal. These data can be found in Fig. 1h and distribution within individuals has been included in **Supp. Fig. 1h**. Methods have now been updated to reflect this change.*

This statement “This overall analysis of ChP translation protein synthesis revealed meaningful differences between the dark and light phases in a Bmal1-dependent manner, with higher levels of protein synthesis during the dark phase” is unfounded. Authors provide no experimental evidence that this is dependent on Bmal1- ie., Bmal1 KO experiment.

Thank you for bringing this imprecise wording to our attention. We have now ungrouped these statements and reported our findings—that translation and protein synthesis in ChP are higher during the dark phase and that the rhythm of pS6 levels are dependent on Bmal1.

Line 197- please indicate that supp fig 3j is the liver to be clear.

We have updated this line to be explicit about the data in Supplemental Figure 3j.

Figure 3m and supp fig 3l- it is unclear why different timepoints were chosen here. Previously 9AM and 9PM were being compared and now 5AM-11AM and 5PM-11PM. Authors should at least indicate the reasoning for this.

Thank you for encouraging us to explain our reasoning more fully. These timepoints were chosen to more closely interrogate the time during which the TTR transition occurs— between 6pm and midnight (Fig. 3k). We have now updated the description of these results.

Line 321-322: Slc9a8 and Nrf2 mentioned to be in fig 6a-b, S6a are not- were these not measured? Or not significant in this dataset compared to previously reported studies references? Adding this information would be useful.

Thank you for pointing out these typographical errors. The appropriate gene name for LAT2 is Slc7a8. This has been corrected. We had previously neglected to list the zinc transporters that were found to be changed (Slc39a9, Slc30a5, Slc30a1). These gene names have now been added. The reference to Nrf2 was as a downstream target of zinc transporters in kidney, but this detail proved to be distracting from the discussion, so we simplified the statement.

Fig S6a: Itgb8 is presented twice in colour map.

*We have clarified our reasoning and reporting for this duplication. The heatmap is grouped by functional class, and therefore some genes are reported multiple times, like Itgb8, which is classified as a CAM and is critical for focal adhesion. Since this division isn't clear in the original single heatmap, heatmaps in **Supp. Fig. 6a** are now displayed as separate maps divided by functional class.*

Reviewer #4 (Remarks to the Author):

This study identified differentially expressed genes during diurnal changes by ribosome profiling of ChP epithelial cells. With identified genes, the authors identified several biological processes that related to diurnal variations in mouse ChP and CSF. The findings can be very important to this field. However, there are many concerns need to be addressed:

We thank the reviewer for their careful analysis of our manuscript, confidence of the importance of this work to the field, and suggestions to improve the manuscript. Individual comments are addressed below.

Major concerns:

1. On line 107-109: the author over claimed that “Overall, translation was increased during the dark phase and diurnal variation in translation was dependent on systemic Bmal1 expression.” based on the data they presented on Fig1. They need to do more to make the conclusion solid, for example, Per2 expression, nucleolar volume in LV ChP of Bmal1 ^{-/-} mice.

Thank you for bringing this imprecise wording to our attention. We have now ungrouped these statements and reported our findings more clearly— that translation and protein synthesis in ChP are higher during the dark phase and that pS6 levels are dependent on Bmal1. Per2 oscillation is already known to be dependent on Bmal1 expression (e.g. Koronowski, et al Cell 2019).

2. The author claimed that choroid plexus translation is diurnally regulated. Ribosome biogenesis is one readout for increased translation capacity and any gene related to ribosome may affect ribosome biogenesis. Unfortunately, Rpl10a is one of the ribosome genes and TRAP mouse line is based on Cre-dependent overexpression of Rpl10a from Rosa26 locus. The author should have avoided TRAP mouse line to profile gene expression. Besides, the genotype of L10aGFP (homozygous or heterozygous) in the Foxj1-Cre::L10aGFP line is also expected to affect ribosome biogenesis since it can be one allele overexpression of Rpl10a or both-alleles overexpression of Rpl10a. Another mouse line, RPL22ha(Ribotag), can be a better choice, because the Rpl22-HA expression is much more “endogenous”.

We thank the reviewer for suggesting a discussion on this topic. Firstly, we have updated the Methods and the Results sections address the important metrics that the mice used for all timepoints were heterozygous for L10aGFP and heterozygous for Foxj1::Cre. Therefore, any effects of the transgene would be consistent across timepoints. Further, TRAP mice have been used in hundreds of studies, and these studies have not discovered altered ribosome biogenesis. In variable systems of over-expression (e.g. CMV-promoter based viral expression) such changes in overall ribosome biogenesis might be a consideration, but not likely with our heterozygous Cre approach. Secondly, our analysis of ribosome biogenesis using nucleolar labeling (Fig. 1 g-h) was performed on WT mice, and therefore this finding is independently verified in mice without the TRAP expression.

3. on line 129-131: “Analyses of all ChP transcripts associated with ribosomes revealed 779 differentially translated transcripts at 9 a.m. vs. 9 p.m.: 431 enriched at 9 a.m., and 348 enriched at 9 p.m.” However, from figure 2a, I realize that the sample 3 from group “9AM” is grouped with sample 1 from group “9PM”, also, even in the same group, there are big variations. The data quality is worrisome. To make the data solid, the author may consider adding more “N”.

Thank you for suggesting further bioinformatics analyses. In general, a few outliers sometimes have the ability to skew hierarchical clustering without reducing the meaningful data gleaned from such analyses. However, after considering this possibility with a bioinformatician, we decided that z-scores (which describes a value in relationship to the mean) are more meaningful to use for hierarchical clustering of large datasets rather than solely log₂[fold change] as had been done in the original submission. Using this more appropriate metric of z-scores, all three of the 9 AM samples cluster and all three of the 9 PM samples cluster (Fig. 2a). Thanks to his additional analysis, our confidence in the TRAP sequencing results is further boosted, in addition

to the support provided by the subsequent functional validation we perform in the rest of the study.

4. To pick top differentially expressed genes, it would be better to use a volcano plot.

*We have now included a volcano plot to additionally represent differentially expressed genes. The full volcano plot is in **Supp Fig. 2a** and a volcano plot has replaced the heatmap in **Fig. 2e**.*

5. Figure 6b, all the gene expression is at transcription level, not protein level, also, many of them show big variations.

We now explicitly state that these data in Figure 6b are transcripts not protein, to differentiate from previous protein-based analyses in the study. While we recognize variability in these data, statistical analyses of rhythmicity were performed to take this into account and indicate rhythmicity in gene expression for these barrier-associated components. Importantly, the rhythmicity in transcript level is striking in so much as it does not fully explain the ribosomal enrichment of these barrier associated proteins. Indeed, the more striking day-night variation at the protein level further supports our use of ribosomal pulldown and highlights that future investigations of ChP circadian regulation should not rely exclusively on mRNA levels. We have now further emphasized this point for readers.

Figure 6g-h, three samples per group is too few. The data is not convincing enough, need more N.

We have now further clarified how these measurements were made. Each point graphed in Figure 6h is representative of an average of 10 junctions from a distinct animal, and each junction was measured 10 times along its apical-to-basal expanse so as to most appropriately represent the width of these junctions. You can see measurements from each junction in Supplementary Figure 6—the widest values are not found in the 9pm cohort, suggesting closer junctions at that time. Because each point represents a biological replicate (one animal) and is an average of repeated measures (10 distinct junctions) within an N, the appropriate error bars were used (SEM, standard error of the mean). The variances within each group are not significantly different ($p=0.17$). The difference between the means is 11.67 ± 3.969 (mean AM = 31.57 nm; mean PM = 19.91 nm). As previously reported, the p-value equals 0.0424, ($p(x \leq T) = 0.9788$). Therefore, the chance of type I error (rejecting a correct H_0) is small: 0.04248 (4.25%). In other words, the difference between the sample average of the AM vs PM groups is big enough to be statistically significant.

We disagree with the reviewer's conclusion that the power of our study is insufficient. To support our reasoning, we have performed a post-hoc power analysis (<https://www.statskingdom.com>). With the variability in our data (AM SD=6.57; PM SD=2.01), a significant difference can be claimed with N=3 in each group. The test power is 0.84 (greater than the generally accepted minimum power of 0.8) as shown with a Normal T- statistical power test using $\alpha \leq 0.05$ and an effect size (d) of 2.41.

Cohen's Effect Size (d) calculation

$$s^2 = \frac{s_1^2 + s_2^2}{2} \qquad s^2 = \frac{6.57^2 + 2.01^2}{2} \qquad s = 4.85824$$

$$d = \frac{|\bar{x}_1 - \bar{x}_2|}{s} \qquad d = \frac{|31.6 - 19.9|}{4.8582} \qquad d = 2.4083$$

The effect size for this analysis (d = 2.41) was found to exceed Cohen's (1988) convention for a large effect (d = 0.8).

Minors:

1. On line 57: "The choroid plexus (ChP) is a key source of CSF. The specialized ChP epithelial cells that": the author should had given the full name of ChP in the ABSTRACT.

We have now updated the abstract to define ChP as choroid plexus.

2. Figure 1d, the author need to show the first timepoint "mid" better, it looks like the "mid" data was plot on Y-axis.

This graph (Fig 1d) and similar ones (Supp Fig1 c-d) have been updated so that the data collected at midnight is no longer plotted on the y-axis.

REVIEWERS' COMMENTS

Reviewer #1 (Remarks to the Author):

The authors have adequately addressed the minor concerns that I raised with the previous submission in the revised manuscript. I have no further concerns about the revised manuscript. This is a beautiful work.

Reviewer #3 (Remarks to the Author):

Review of revised manuscript Fame et al.

Thank you to the authors for revising the manuscript and clarifying the data and interpretation. This is an exciting and important body of work.

VERY minor edit below:

Line 152 enriches-should be "enriched".

Reviewer #4 (Remarks to the Author):

The authors have successfully addressed all of my concerns, resulting in significant improvements to the manuscript and making it more suitable for publication.

REVIEWERS' COMMENTS

Reviewer #1 (Remarks to the Author):

The authors have adequately addressed the minor concerns that I raised with the previous submission in the revised manuscript. I have no further concerns about the revised manuscript. This is a beautiful work.

We thank the reviewer for their careful reading of our revised manuscript and enthusiastic support for its publication.

Reviewer #3 (Remarks to the Author):

Review of revised manuscript Fame et al.

Thank you to the authors for revising the manuscript and clarifying the data and interpretation. This is an exciting and important body of work.

VERY minor edit below:

Line 152 enriches-should be "enriched".

This typo has now been corrected. Our thanks to the reviewer for this additional helpful comment and for their strong endorsement of this revised work.

Reviewer #4 (Remarks to the Author):

The authors have successfully addressed all of my concerns, resulting in significant improvements to the manuscript and making it more suitable for publication.

We are grateful to the reviewer for their thorough review of our revised manuscript and for their positive assessment of our work.